# Proton-assisted calcium-ion storage in aromatic organic molecular crystal with coplanar stacked structure

Cuiping Han [1], Hongfei Li [2✉], Yu Li[3,4], Jiaxiong Zhu[2] & Chunyi Zhi [1✉]

Rechargeable calcium-ion batteries are intriguing alternatives for use as post-lithium-ion batteries. However, the high charge density of divalent $Ca^{2+}$ establishes a strong electrostatic interaction with the hosting lattice, which results in low-capacity Ca-ion storage. The ionic radius of $Ca^{2+}$ further leads to sluggish ionic diffusion, hindering high-rate capability performances. Here, we report 5,7,12,14-pentacenetetrone (PT) as an organic crystal electrode active material for aqueous Ca-ion storage. The weak π-π stacked layers of the PT molecules render a flexible and robust structure suitable for Ca-ion storage. In addition, the channels within the PT crystal provide efficient pathways for fast ionic diffusion. The PT anode exhibits large specific capacity (150.5 mAh $g^{-1}$ at 5 A $g^{-1}$), high-rate capability (86.1 mAh $g^{-1}$ at 100 A $g^{-1}$) and favorable low-temperature performances. A mechanistic study identifies proton-assisted uptake/removal of $Ca^{2+}$ in PT during cycling. First principle calculations suggest that the Ca ions tend to stay in the interstitial space of the PT channels and are stabilized by carbonyls from adjacent PT molecules. Finally, pairing with a high-voltage positive electrode, a full aqueous Ca-ion cell is assembled and tested.

[1] Department of Materials Science and Engineering, City University of Hong Kong, Kowloon, Hong Kong, China. [2] Songshan Lake Materials Laboratory, Dongguan, Guangdong, China. [3] College of Materials Science and Engineering, Shenzhen University, Shenzhen, China. [4] Shenzhen Key Laboratory of Special Functional Materials, Shenzhen University, Shenzhen, China. ✉email: lihf@sslab.org.cn; cy.zhi@cityu.edu.hk

Recently, a flurry of research activity has been devoted to developing alternative battery technologies in addition to lithium-ion batteries (LIBs) based on earth-abundant elements[1–4]. Rechargeable calcium-ion batteries (CIBs) can offer considerable advantages including cost-effectiveness and high volumetric/gravimetric capacity (2072 mAh mL$^{-1}$ and 1337 mAh g$^{-1}$)[5] due to their high abundance and a relatively lower reduction potential (i.e., −2.87 V, −2.37 V, and −1.66 V vs. SHE for Ca/Ca$^{2+}$, Mg/Mg$^{2+}$, and Al/Al$^{3+}$, respectively)[6] and the divalent electron redox properties of calcium[7]. However, current research on CIBs is still in its infancy because the reversible plating/stripping of metallic calcium is possible only with judiciously tailored nonaqueous electrolytes[8–11]. Moreover, severe undesired side reactions accompany the deposition of calcium, such as the formation of CaF$_2$, CaCl$_2$, CaCO$_3$, and CaH$_2$, leading to low Columbic efficiency and continuous growth of a passivation film on the calcium electrode surface[8,12,13]. The complexity of working directly with a Ca-metal anode drives the need for alternative anode materials. Another challenge of CIBs, both nonaqueous and aqueous, is the absence of high-performance electrode materials. The high charge density and the relatively large ionic radius of divalent Ca$^{2+}$ (0.099 nm) make these ions interact more strongly with the hosting lattice than monovalent cations, which results in sluggish Ca$^{2+}$ ion diffusion in inorganic crystals[14]. For example, Rong et al. investigated the migration of several multivalent ions (Mg$^{2+}$, Zn$^{2+}$, Ca$^{2+}$, and Al$^{3+}$) in spinel Mn$_2$O$_4$, olivine FePO$_4$, layered NiO$_2$, and orthorhombic δ-V$_2$O$_5$ and found that the mobility of multivalent ions is consistently lower than that of Li$^+$. Furthermore, the barrier of different bivalent ions depends greatly on the hosting structure[15]. High multivalent ion mobility in solids is only possible by judicious tuning of crystal structure and chemistry[16]. However, to date, only a handful of anode materials have been demonstrated, and the best-known anode materials are Sn-based alloys[17,18] and graphite[19]. Their performances, especially the specific capacity and cycling stability, are still far from satisfactory. The specific capacities of the reported CIBs fade rapidly to the best of our knowledge, and the cycling life barely exceeds 1000 cycles. Therefore, the exploration of electrode candidates capable of reversible Ca$^{2+}$ ion storage is critically important for the development of CIBs.

Advantageously, organic solids are assembled by weak van der Waals forces, which endow them with a more flexible solid structure and low repulsion for Ca ion diffusion. Moreover, organic materials show the merits of high abundance, high electrochemical reactivity, and large structural diversity. Thus, they are promising candidate electrodes for electrochemical energy storage applications[20–22]. Unfortunately, little attention has been given to the exploration of Ca$^{2+}$-hosting organic electrodes, and reports of their performance is very limited[23–26]. Organic solids with planar structures and efficient diffusion pathways are highly favorable for facilitating the storage and diffusion of charge-dense and bulky Ca$^{2+}$ ions. In this regard, aromatic organic crystals, made of molecules that contain benzene ring subunits, are potential electrodes for high-rate CIBs due to their readily formed channels and voids in the molecules. To date, the organic aromatic molecule family of crystals has remained unexplored for CIBs. More importantly, little understanding has been gained on the storage chemistry or structural evolution of organic materials during multivalent ion storage, including Ca$^{2+}$ storage. Therefore, it is vital to search for high-performance aromatic organic crystals and clarify their detailed charge storage process, including their reactive sites, molecular structural changes and possible side reactions (e.g., the co-insertion of proton or decomposition of the electrolyte solvent).

Here, we report aromatic organic molecular crystals, represented by 5,7,12,14-pentacenetetrone (PT), as a high-rate and long-life CIB anode material in mild aqueous electrolytes. The aromatic PT molecules are bonded with van der Waals forces, forming a coplanar stacked layered structure with rich 1D channels perpendicular to the molecular layers. The flexible structure of organic materials and the presence of 1D channels facilitate the reversible uptake/removal of charge-dense Ca ions, providing a diffusion coefficient on the order of 10$^{-8}$–10$^{-11}$ S cm$^{-1}$. As expected, the PT anode exhibits promising Ca ion storage performance, including a high discharge capacity of 150.5 mA h g$^{-1}$ at 5 A g$^{-1}$, superior retained rate capacity of 86.1 mA h g$^{-1}$ at a large specific current of 100 A g$^{-1}$, and good capacity retention at low temperature. A mechanistic study revealed that protons in aqueous electrolytes significantly contribute to the Ca ion storage and that the highly reversible chemical adsorption and desorption of both protons and Ca$^{2+}$ with carbonyl groups comprise the dominant redox reaction[27]. Furthermore, theoretical calculations reveal that the negative charges of Ca-based enolate are delocalized across the stacks due to π–π stacking interactions such that one Ca$^{2+}$ ion is stabilized by four carbonyls from adjacent PT molecules. Finally, an aqueous full Ca-ion cell is prepared by pairing the PT-based electrode with a KCoFe(CN)$_6$•xH$_2$O cathode, rendering a high voltage of 2.1 V and good capacity performance. These findings contribute to understanding the charge storage behavior of organic electrode materials.

## Results and discussion

Several organic aromatic molecular crystals with different numbers of aromatic rings and carbonyl groups at different positions were selected and compared, including 2,5-dichloro-1,4-benzo-quinone (2,5-dichloro BQ), 2,5-dimethoxybenzo-1,4-quinone (DMQ), 1,5-dichloro-anthraquinone (1,5-dichloro AQ), phenanthrenequinone (9,10-PQ), perylene-3,4,9,10-tetracarboxylic dianhydride (PTCDA), and PT. All the selected electrodes are feasible for studying highly reversible Ca$^{2+}$ ion uptake in an aqueous CaCl$_2$ electrolyte. Generally, organic compound redox properties are directly related to their chemical structure and can be tuned through judicious incorporation of appropriate functionalities. Aromatic organic molecules with fewer electron-donating phenyl groups (–C$_6$H$_5$), such as DMQ and 2,5-dichloro BQ, show relatively higher redox potentials (Fig. 1a). However, their capacity is limited to 40–60 mA h g$^{-1}$, most likely due to the high charge density of Ca$^{2+}$, causing strong electrostatic repulsion from the small molecules. PT displays relatively low redox potential and exciting specific capacity values with a large aromatic ring and a high carbonyl group concentration, making it a promising anode material for aqueous CIBs. Structural and morphological characterizations were first determined on the PT material. The XRD profile shows a series of distinct peaks at 2θ = 8°, 12°, 13.4°, 15.4°, 20.8°, 22.2°, 24.3°, and 28.2°, with d spacing of 1.1, 0.74, 0.65, 0.57, 0.43, 0.40, 0.37, and 0.32 nm, respectively (Fig. 1b). The presence of strong diffraction peaks confirms the highly crystalline nature of PT as an aromatic organic molecular crystal, which is different from conventional amorphous organic materials. Rietveld refinement was then applied to the XRD profile, and a triclinic lattice with P$\bar{1}$ symmetry and cell parameters of a = 0.476 nm, b = 0.754 nm, c = 1.112 nm, α = 97.68°, β = 93.23°, and γ = 99.13° was obtained (Supplementary Table 1). According to the refined crystal structure, all PT molecules are uniformly aligned, forming a coplanar packed motif (Fig. 1c)[28]. Perpendicular to this coplanar direction, the PT molecules are π–π stacked layer-by-layer, forming 1D molecular channels (pink circles in Fig. 1c) that are walled by the adjacent PT molecules.

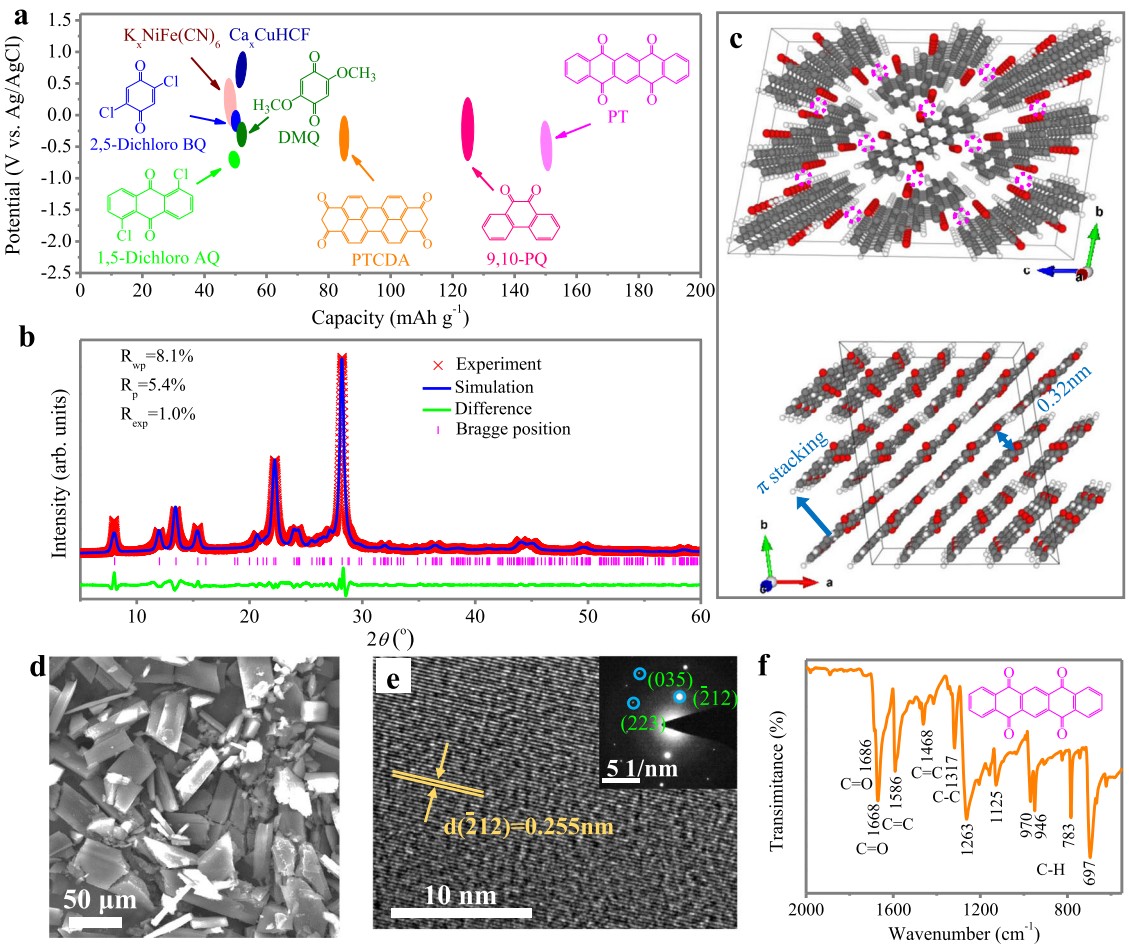

**Fig. 1 Comparison of selected aromatic organic molecular crystals and structural characterization of the PT materials. a** Discharge/charge voltage ranges and capacities of selected aromatic organic molecular crystals (DMQ, 2,5-dichloro BQ, 1,5-dichloro AQ, 9,10-PQ, PTCDA, and PT in 1M CaCl₂ electrolyte. The typically reported inorganic electrodes including Ca$_x$CuHCF[32], K$_x$NiFe(CN)$_6$[33], are also listed for comparison. **b** XRD pattern of PT powder (red cross) and the Rietveld-refined XRD pattern (blue line). **c** Perspective view of a PT supercell observed from a axis and c axis. **d** SEM image. **e** HRTEM image and SAED pattern. **f** FTIR spectrum of PT materials.

The existence of these 1D channels is expected to facilitate the diffusion of Ca ions within the PT crystal. The interplanar distance is ~0.32 nm, which is very close to the interlayer spacing in graphite. The morphological observation reveals that pure PT materials are microsized rods with 50–80 μm in length and 20–40 μm in width (Fig. 1d). High-resolution transition electron microscopy was also used to confirm the crystalline nature. Lattice fringes of the ($\bar{2}$12) space group with a *d* spacing of 0.255 nm and the corresponding selected-area electron diffraction (SAED) patterns were observed (Fig. 1e). Then, the functional groups of PT were revealed by Fourier transform infrared (FTIR) spectroscopy (Fig. 1f). Specifically, characteristic peaks at approximately 1686 and 1668 cm⁻¹ can be assigned to the stretching vibration of carbonyl groups[29,30]. The strong adsorption bands at 1586 and 1468 cm⁻¹ represent the stretching vibration of C=C bonds in the aromatic ring[30]. The peak at 1317 cm⁻¹ is assigned to the C–C stretching vibration. The peaks at 1263 and 1125 cm⁻¹ can be attributed to C–H in-plane bending vibrations while the peaks at 970, 946, 783, and 697 cm⁻¹ are attributed to C–H out-of-plane bending vibrations in the aromatic ring[31].

To evaluate the electrochemical property of the PT crystal as Ca²⁺ ion-hosting materials, cyclic voltammetry (CV) profiles were first recorded at 20 mV s⁻¹ in 1 M CaCl₂ aqueous solution using three-electrode configuration (Fig. 2a, Supplementary Fig. 1). The initial CV curve of the PT anode displays two

reduction peaks at −0.3 V, −0.66 V vs. Ag/AgCl and three oxidization peaks at −0.47 V, −0.38 V, and −0.2 V vs. Ag/AgCl under 20 mV s⁻¹, respectively. The presence of multiple peaks suggests stepwise and multiple electron transfer during the Ca²⁺ insertion and extraction process. Interestingly, the subsequent CV curves are slightly different from the initial curve. The cathodic peak at −0.3 V vs. Ag/AgCl disappears in the following cycles, due to the reduction of dissolved O₂ in the electrolyte, as shown in Supplementary Fig. 2. Moreover, the reduction peak at −0.66 V slightly shifts to −0.63 V and is gradually amplified with increasing cycle number. In addition, a shoulder reduction peak at −0.85 V progressively emerged. In the anodic process, the gradual increase in peak intensity and the presence of an additional shoulder peak at −0.76 V were also observed, all suggesting the activation of PT during repeated Ca²⁺ storage. This activation process is most likely related to the morphological evolution of large PT bulks to small nanowires as discussed in a subsequent section, leading to the gradual exposure of inner active surface but also contributes to fast ion diffusion and efficient charge transfer due to better accessibility for charge carriers infiltrating into the cathode materials. The origin of each peak is discussed in a subsequent section.

The following galvanostatic charge/discharge (GCD) test further revealed the promising electrochemical performance of the PT anode. At an initial specific current of 5 A g⁻¹, a gradual

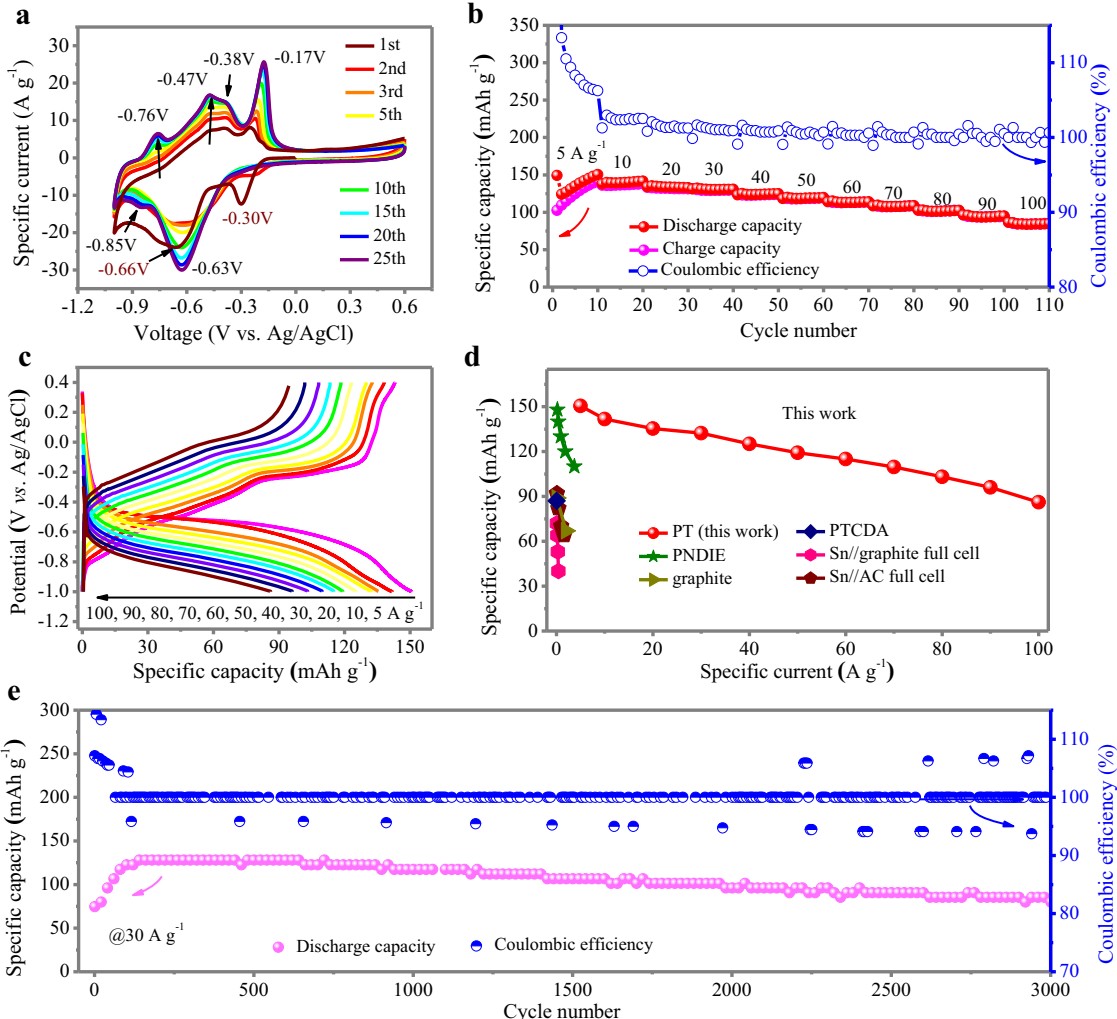

**Fig. 2 Electrochemical performance of the PT anode in a three-electrode cell. a** CV curves of PT anode in 1M CaCl$_2$ solutions at 20 mV s$^{-1}$. **b** Specific capacities of PT in 1M CaCl$_2$ at different specific currents and the corresponding Coulombic efficiencies. **c** GCD curves of PT in 1M CaCl$_2$ at different specific currents. **d** Capacity and rate ability comparison with reported anodes for CIBs. See Supplementary Table 2 for reported data in literature. **e** Cycling stability and Coulombic efficiency of PT in 1M CaCl$_2$ at 30 A g$^{-1}$.

increase in specific capacity was observed due to the activation process (Fig. 2b). It achieved a high specific discharge capacity of 150.5 mAh g$^{-1}$ during the 10th cycle, corresponding to roughly one-half of its theoretical value (317 mAh g$^{-1}$, based on the reduction of four carbonyls). Therefore, it is reasonable to infer that one-half the carbonyl groups in the PT crystal are utilized during Ca ion storage because the insertion of large amounts of Ca$^{2+}$ ions induces strong electronic repulsion within the crystal. With increasing specific currents from 10 to 20, 30, 40, 50, 60, 70, 80, and 90 A g$^{-1}$, the capacities are effectively retained at 141.7, 135.4, 132.3, 125.1, 119.2, 115, 109.7, 103.1, and 96 mAh g$^{-1}$, respectively, showing an impressive capacity retention rate. Notably, the PT anode can be charged and discharged at high specific currents, up to 100 A g$^{-1}$. The PT anode takes only 6.2 s to complete a charge–discharge cycle at this specific current, while maintaining an appealing discharge capacity of 86.1 mAh g$^{-1}$ (Fig. 2b), showing a supercapacitor-level high-rate capability. The GCD profiles during the 10th cycle of each specific current exhibit a discernible plateau during discharging and a sloped line in combination with a plateau during charging (Fig. 2c), which were consistent with the shape of the CV curves in Fig. 2a. High Coulombic efficiency of >100% was achieved, mainly at 5 A g$^{-1}$, as shown in Fig. 2b, which is most likely related to the activation

process of the PT anode. At the beginning of the GCD test, the activation process was not completed, and the current was not fully applied to the PT material, which led to some side reactions, such as the hydrogen evolution reaction (HER), which was the main reason for the high Coulombic efficiencies in the first few cycles.

The demonstrated capacity values and rate capability surpass the currently reported anode materials for CIBs (Fig. 2d, Supplementary Table 2), such as PTCDA (87 mAh g$^{-1}$ @ 0.02 A g$^{-1}$)[23], poly[N,N′-(ethane-1,2-diyl)-1,4,5,8-naphthalenetetracarboxiimide] (PNDIE) (148 mAh g$^{-1}$ @ 1C with 1C = 183 mA g$^{-1}$)[24], Sn anode in organic CIBs[17,18] and graphite[19], representing the highest level for CIBs reported thus far. More importantly, organic materials generally suffer from rapid capacity fading due to high solubility in aqueous electrolytes. Here, in our case, the insoluble nature of PT in water renders the as-prepared PT anode with a robust cycle life of 3000 cycles at 30 A g$^{-1}$ (Fig. 2e). Initially, the PT material delivered a specific capacity of 74.7 mAh g$^{-1}$ at 30 A g$^{-1}$. After a gradual activation process, a maximum discharge capacity of 128 mAh g$^{-1}$ was achieved, which remained at 85.3 mAh g$^{-1}$ after 3000 cycles. The activation process at the initial cycles matches well with that depicted in Fig. 2a, b.

To explore the mechanism for such a high-rate capacity of the PT anode, the electrochemical reaction kinetics were analysed by

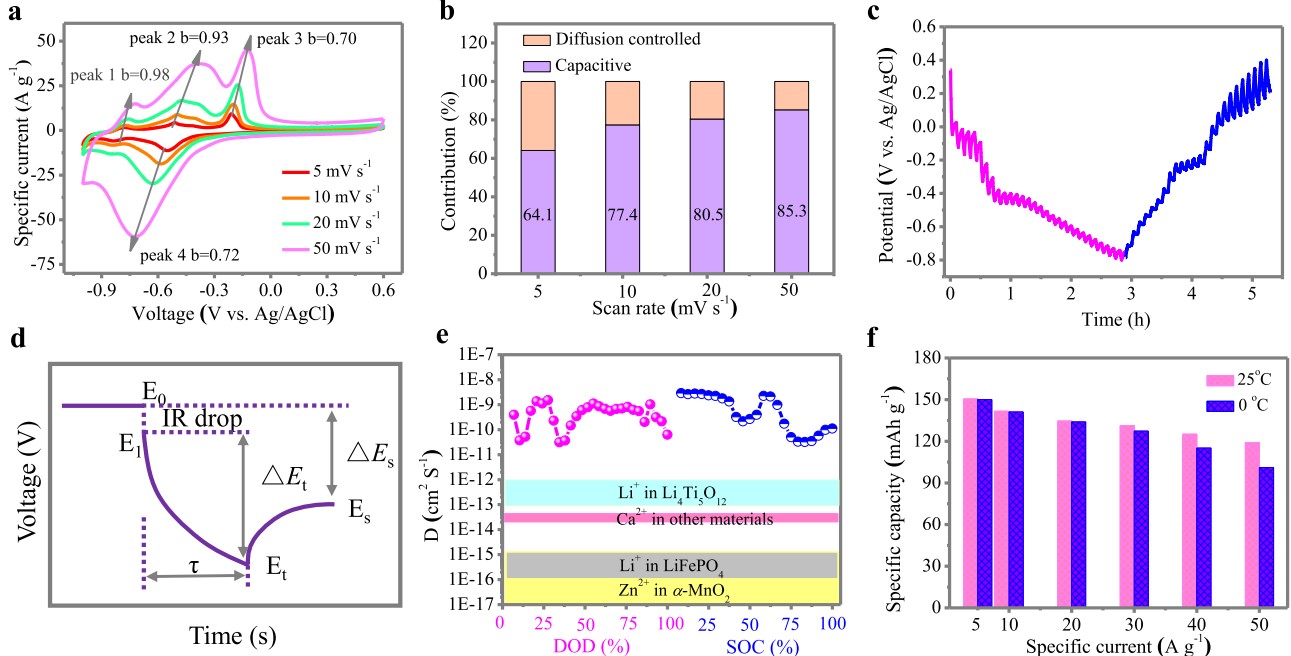

**Fig. 3 Reaction kinetics of the PT anode in a three-electrode cell. a** CV plots of PT anode in 1M CaCl$_2$ at scanning rates from 5 to 50 mV s$^{-1}$. **b** Contribution ratio of the capacitive- and diffusion-controlled process at various scan rates. **c** GITT curves of PT anode during discharge and charge. **d** The key parameters for diffusion coefficient calculation. **e** The calculated Ca$^{2+}$ ion diffusion coefficient during discharging and charging process and comparison with other systems. **f** Specific capacities of PT in 1M CaCl$_2$ at room temperature and 0 °C.

recording the CV curves at various scan rates. With an increasing scan rate from 5 to 50 mV s$^{-1}$, the CV curves displayed similar shapes with gradually magnified specific currents (Fig. 3a). In principle, the current response of an electrode material can be correlated to the scan rate according to Eq. 1[34,35]:

$$i = a v^b \qquad (1)$$

Where a and b are constants. By plotting log ($i$) vs. log ($v$) (Supplementary Fig. 3), the b values for the four marked redox peaks in Fig. 3a are calculated to be 0.98, 0.93, 0.70, and 0.72. The distinct b values indicate varied reaction kinetics for each peak. The capacitive contribution at each scan rate is further quantified according to previous reports[36,37]. At 10 mV s$^{-1}$, the capacitive effect contributed 77.4% of the total capacity (i.e., the shaded area in Supplementary Fig. 4). Moreover, the contribution ratio increases progressively with the scan rate. It plays a leading role in the total capacity at high scan rates of 50 mV s$^{-1}$ (Fig. 3b), signifying a high-power capability of PT materials due to their flexible structure with 1D channels for efficient Ca$^{2+}$ ion diffusion.

The effective diffusion of Ca$^{2+}$ ions in organic PT crystals, which is a great challenge for CIBs due to the high charge density and the bulk size of Ca$^{2+}$, was quantified using the galvanostatic intermittence titration technique (GITT)[38,39]. A series of current pulses followed by a relaxation process was applied to the PT anodes (Fig. 3c). The chemical diffusion coefficient (D) of Ca$^{2+}$ in the active material can be estimated according to Fig. 3d and the following Eq. 2[38,39]:

$$D = \frac{4}{\pi \tau} \left( \frac{m V_m}{MS} \right)^2 \left( \frac{\triangle E_s}{\triangle E_\tau} \right)^2 \qquad (2)$$

Where $\tau$ is the constant current pulse duration (s); $m$, $M$, and $V_M$ are the mass (g), molar mass (338.3 g mol$^{-1}$) and molar volume (222.9 cm$^3$ mol$^{-1}$) of PT, respectively; S is the electrode-electrolyte interface area (taken as the geometric area of the

electrode, ~1 cm$^2$); $\triangle E_S$ is the steady-state potential change (V) by the current pulse, and $\triangle E_t$ is the potential change (V) during the constant current pulse after eliminating the iR drop.

The calculated diffusion coefficient of the Ca$^{2+}$ ions ($D$) in the PT anode during discharging and charging is in the range of $10^{-8}$–$10^{-11}$ cm$^2$ S$^{-1}$ (Fig. 3e), which is significantly higher than that of Li$^+$ in LiFePO$_4$ ($10^{-14}$–$10^{-16}$ cm$^2$ S$^{-1}$ as determined by GITT[40]) and Li$_4$Ti$_5$O$_{12}$ ($10^{-11}$–$10^{-12}$ cm$^2$ S$^{-1}$ as determined by potentiostatic intermittent titration technique[41] and $10^{-12}$–$10^{-13}$ cm$^2$ S$^{-1}$ as determined by CV[42], Zn$^{2+}$ in α-MnO$_2$ ($10^{-15}$–$10^{-17}$ cm$^2$ S$^{-1}$ as determined by GITT[43], Ca$^{2+}$ in other materials (e.g., Ca$^{2+}$ in Ni-based metal-organic compound, 5.3 × $10^{-14}$ cm$^2$ S$^{-1}$ as determined by electrochemical impedance spectroscopy)[44]. The high apparent diffusion coefficient of Ca$^{2+}$ ions ($D$) in the PT anode can be ascribed to its weakly stacked layered structure with 1D molecular channels, which provides an efficient Ca$^{2+}$ ion diffusion pathway, thus leading to enhanced reaction kinetics of PT during the electrochemical reaction process.

To gain a comprehensive understanding of the electrochemical behavior of the PT material, the capacities of PT measured at 0 °C and room temperature (~25 °C) were compared (Fig. 3f). According to the Arrhenius equation, achieving superior electrochemical performance at low temperatures is very difficult since the reaction rate is significantly reduced[45]. Notably, the PT anode underwent only very slight capacity degradation, with 84.8% capacity retention at a high specific current of 50 A g$^{-1}$, signifying good reaction kinetics of this aromatic organic crystal in the low-temperature range.

We now turn our attention to scrutinizing the detailed charge storage chemistry of PT crystals in aqueous solutions. For aqueous zinc-based batteries, the co-insertion of protons (H$^+$ or H$_3$O$^+$) with Zn$^{2+}$ is widely reported due to the smaller size of protons[27,46–48]. However, the role of protons during the storage of Ca$^{2+}$ remains undiscernible in aqueous CIBs. To clarify the effect of protons in this CIB system, the CV curve of the PT anode with

                                                                                                                                5

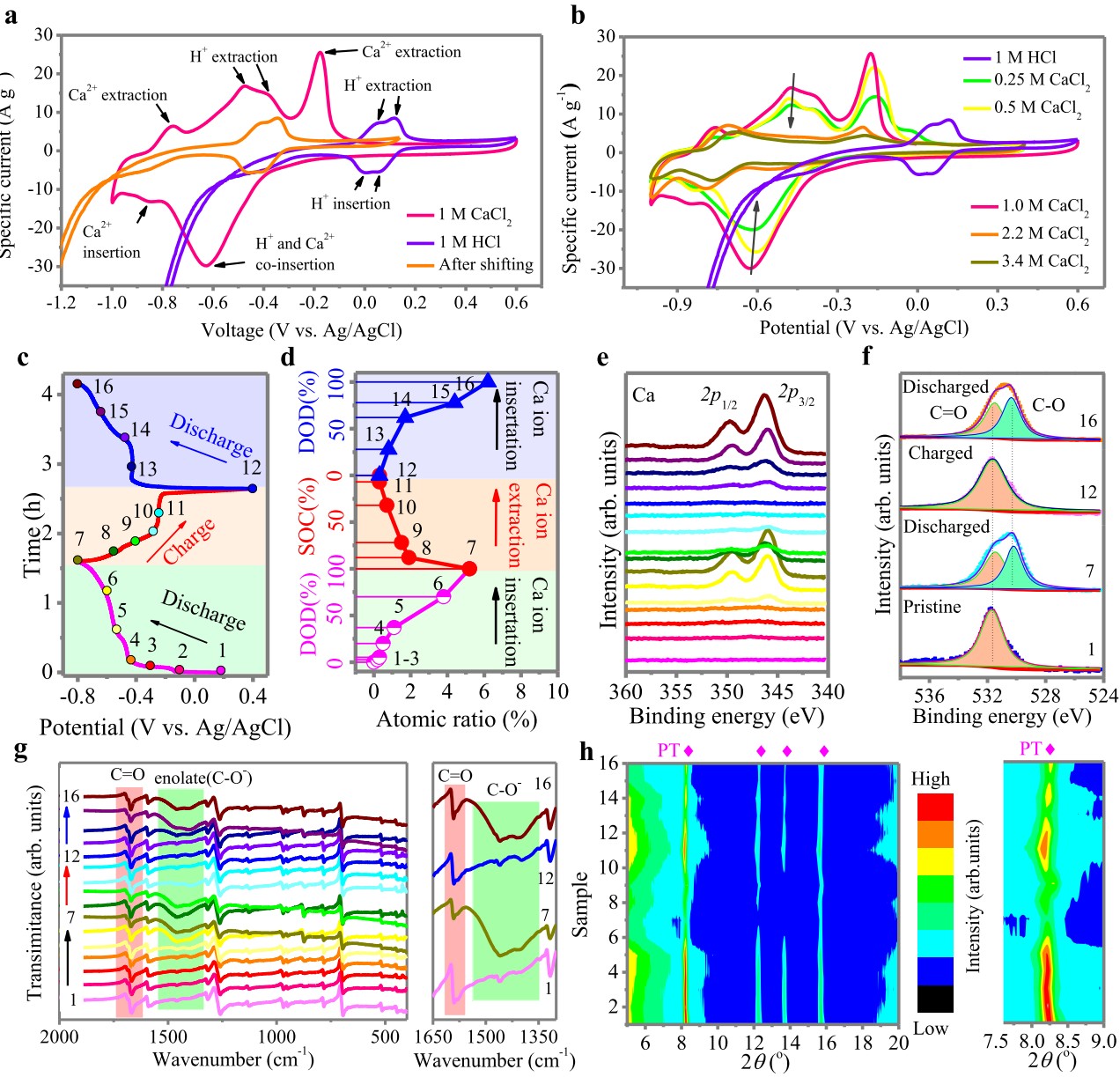

**Fig. 4 Mechanistic study on the Ca-ion storage in PT anode. a** CV curves of PT anode in 1M CaCl₂ and 1M HCl at 20 mV s⁻¹. The orange line represents corresponding CV curves in a pH = 7.8 solution, calculated based on the Nernst equation. **b** CV curves of PT anode in different CaCl₂ solutions and in 1M HCl at 20 mV s⁻¹. **c** Initial discharge-charge curve and the second discharge curve of PT anode under a specific current of 0.1 A g⁻¹. Various ex situ tests were conducted at the marked points. **d** Atomic concentration of Ca at marked points by XPS. **e** Ex situ XPS spectrum of Ca 2p. **f** O 1s XPS spectra at selected points. **g** Ex situ FTIR. **h** Ex situ XRD.

1 M CaCl₂ electrolyte (pH = 7.8) was compared with that in highly concentrated H⁺ solutions (i.e., 1 M HCl electrolyte, pH = 1.0) (Fig. 4a, Supplementary Fig. 5a). In the HCl electrolyte, the PT anode presents two reduction peaks at 0.06 V and 0 V vs. Ag/AgCl and two oxidization peaks at 0.05 V and 0.12 V vs. Ag/AgCl, respectively, which is related to proton uptake/release with the carbonyl groups of PT. The corresponding CV curves for a pH = 7.8 solution (the same pH value as 1 M CaCl₂ aqueous solution) can be obtained by shifting the CV curves measured for the PT anode in 1 M HCl according to the Nernst equation (Supplementary Methods for Calculation based on Nernst Equation)[30]. The location of the two oxidization peaks after shifting partially overlapped with the oxidization peaks at −0.47 V and −0.38 V vs. Ag/AgCl measured in 1 M CaCl₂ electrolyte, suggesting that the proton participated in the storage process of Ca²⁺ ions in the PT

anode. More importantly, proton participation is also observed in similar aromatic organic molecular crystals, such as PQ (Supplementary Fig. 6), suggesting that the findings of this work apply to other aromatic organic materials. By comparing the area under the two oxidization peaks at −0.47 V and −0.38 V vs. Ag/AgCl with the total anodic scan, it is calculated that ~49% of the total capacity is contributed by proton uptake. Moreover, the specific capacity of the PT anode in 1 M HCl was also measured (Supplementary Fig. 7), corresponding to ~43% of the 150.5 mAh g⁻¹ achieved in 1 M CaCl₂. These results collectively suggest that ~40–50% of the total capacity comes from proton uptake.

During the reduction process, H⁺ seems to be inserted before or during the insertion of Ca²⁺ since only two reduction peaks are observed (i.e., −0.85 V and −0.63 V vs. Ag/AgCl) and the insertion potential of H⁺ after shifting (i.e., −0.46 V vs. Ag/AgCl)

is on the right shoulder of the first reduction peak. This supposition was verified by comparing the CV curves of the PT anode measured in the $CaCl_2$ electrolyte of different concentrations (Fig. 4b). As the concentration of $CaCl_2$ electrolyte increases from 1 M to 3.4 M, a significant decrease in the current response in the two oxidization peaks at −0.47 V, −0.38 V vs. Ag/AgCl, and a reduction peak at −0.63 V vs. Ag/AgCl is observed, mainly due to the decrease in proton numbers in the highly concentrated electrolyte. This result verifies that the oxidization peaks at −0.47 V and −0.38 V vs. Ag/AgCl correspond to proton extraction. Furthermore, without the contribution from protons, the insertion of $Ca^{2+}$ negligibly proceeds as indicated by the sharp decline in the reduction peak at −0.63 V. Therefore, the reduction peak at −0.63 V contains a significant proportion of proton insertion. This claim was further confirmed by measuring the CV curves the PT anode in ethylene glycol (EG)-based organic electrolytes (Supplementary Fig. 5b). The PT anode shows no pronounced redox peaks in the EG-based 1 M $CaCl_2$ solution due to the absence of proton. In contrast, a significant increase in specific current and the presence of redox peaks are observed after $H_2O$ is added to the nonaqueous electrolyte, confirming the important role of protons during the insertion of $Ca^{2+}$, although the pH value of the $CaCl_2$ solution is slightly alkaline (Supplementary Fig. 5a).

To better understand the structural evolution of PT crystal during the co-insertion of protons and $Ca^{2+}$ ions, various ex situ tests, including FTIR, X-ray photoelectron spectroscopy (XPS), and X-ray diffraction (XRD) were performed with the PT anodes at the initial two charge–discharge cycles, as shown in Fig. 4c. XPS explorations powerfully demonstrate the reversible uptake of $Ca^{2+}$ during discharge and its removal after recharging (Fig. 4d, e). Even after Ar ion etching at 10 and 20 nm off, the discharged PT anode still exhibits in-depth XPS peaks of Ca $2p$ (Supplementary Fig. 8), verifying Ca bulk storage in the PT anodes. Notably, the concentration of Ca ions at the initial stage of the discharge process is very low and begins to increase after the discharge potential falls below −0.46 V vs. Ag/AgCl (point 4) (Fig. 4d), coinciding with the assumption that $Ca^{2+}$ is inserted after proton insertion (i.e., −0.46 V vs. Ag/AgCl after shifting).

In addition, XPS O $1s$ spectra reveal the appearance of the C–O peak (530.3 eV) after discharging and its disappearance after charging (Fig. 4f), which coincides with the sustained coordination of $H^+$ and $Ca^{2+}$ on the carbonyl groups of PT. The invertible conversion process is also verified by ex situ FTIR (Fig. 4g). In the FTIR spectra, upon discharging, the stretching vibration of carbonyl groups (–C=O) at approximately 1668 $cm^{-1}$ is gradually weakened, while an additional and broad band centered at ~1460 $cm^{-1}$ is strengthened, indicating the conversion of carbonyl groups (C=O) to enolate groups (C–O⁻)[23,49]. During the following charging process, the carbonyl group peaks return to their initial state, while the peaks of enolate groups gradually become weaker. These XPS and FTIR results support the chemical adsorption mechanistic assumption, where the cations adsorb to the negatively charged oxygen atoms upon electrochemical reduction of the carbonyl groups and desorb reversibly during reverse oxidation. Moreover, compared with the traditional intercalation-/deintercalation-based energy storage mechanism, this enolation reaction of quinone (–C=O) to quinone salts (–C–O–M) is beneficial for sustaining the high structural stability of PT materials during repeated cycling. As a consequence, the ex situ XRD data reveal that the most intriguing structural change in PT upon discharging is the slight reduction in peak intensity (Fig. 4h) due to morphological changes, as revealed by transmission electron microscopy (TEM) and scanning electron microscopy (SEM) in a subsequent section. Upon decalcification, the peak intensities are partially restored. No extra

peaks are observed. The PT anodes were subjected to 10 discharge/charge cycles at 0.1 A $g^{-1}$ and demonstrate good crystallinity (Supplementary Fig. 9). Furthermore, the XRD profiles of the PT anodes subjected to a high-rate test at 100 A $g^{-1}$ (Supplementary Fig. 10) and cycling test at 30 A $g^{-1}$ were also obtained (Supplementary Fig. 11), which showed well-retained characteristic peaks, indicating the robust structural stability of PT materials during repeated cycling.

The morphology variation of PT during the discharge and charge process is given in Supplementary Figs. 12, 13. The freshly prepared electrode consists of small conductive carbon and a large bulk of PT (Supplementary Fig. 12b). Upon discharge, this bulk PT converts to highly homogeneous fibers of ~1–2 μm in length and ~200 nm in diameter (Supplementary Fig. 12c), signifying a morphology evolution process. When the voltage is discharged to −0.46 V vs. Ag/AgCl (point 4), spheroidal particles with a large dimension of ~500 nm begin to form, matching well with the XPS data showing that the content of Ca begins to increase after point 4. With the discharge process ongoing, an increasing number of spheroidal particles are formed and randomly distributed around the fibers (Supplementary Figs. 12d, g, 13), which are quite different from the initial PT materials. Upon charging, these spheroids gradually disappear, with only fiber-shaped PT remaining in the electrode (Supplementary Fig. 12e, f). The formation and disappearance of these spheroidal particles are highly reversible during the discharge and charge process. A TEM characterization of both fibers and spheroids was conducted (Fig. 5, Supplementary Fig. 14), Surprisingly, Ca was not observed in the fibers as suggested by electron energy-loss spectroscopy (EELS) and TEM-energy-dispersive spectroscopy (EDS) mapping images (Supplementary Fig. 14), but was concentrated on the spheroids formed in the discharged state (Fig. 5a–c), with a weight percent of C:Ca = 76.3:23.7 by EELS. Inductively coupled plasma-mass spectrometry (ICP-MS) analysis was also used to quantify the Ca concentration in the whole PT anode, which was measured to be 16.9 g $kg^{-1}$ (Supplementary Table 3). The high-resolution TEM image reveals the crystalline nature of the Ca-inserted products and lattice fringes with an interlayer space of 0.272 nm (Fig. 5d). Fast Fourier transform was conducted on the selected area (marked by yellow frame), and the yielded diffraction pattern can be well indexed to the triclinic PT phase (Fig. 5e). The corresponding SAED pattern over a broad area displays diffraction rings from the unreacted PT phase (0.32, 0.255, and 0.162 nm) and the Ca-inserted PT phase (0.272 nm) (Fig. 5f), which is consistent with the XRD measurement that the Ca-inserted phase adopts the same crystal structure as fresh PT. Therefore, it is assumed that the PT anode experienced a gradual morphology evolution from the large bulk form to nanofibres and finally to nanosphere aggregates during calcium uptake, but the underlying reason for this morphological evolution needs further investigation.

The accommodation of $Ca^{2+}$ in the PT crystal is quite complicated because of the divalent nature of $Ca^{2+}$, which is not well documented in literature. This is because the storage of one $Ca^{2+}$ is expected to cause the enolation of at least two carbonyl groups. This is very different from monovalent ions (e.g., $Li^+$, $Na^+$, or $K^+$), where one carbonyl group binds one cation. Moreover, there is a high possibility that these carbonyl groups come from adjacent PT molecules to suppress the strong electrostatic interaction. Theoretical calculations based on density functional theory (DFT) were conducted to simulate the possible extent of $Ca^{2+}$ storage in PT crystals. Individual geometries of the PT molecule and its crystal structure were optimized at the Perdew–Burke–Ernzerhof (PBE) functional. The corresponding lattice parameters match well with the experimental values (Supplementary Table 1). Then, the molecular electrostatic potential (MESP) was calculated to

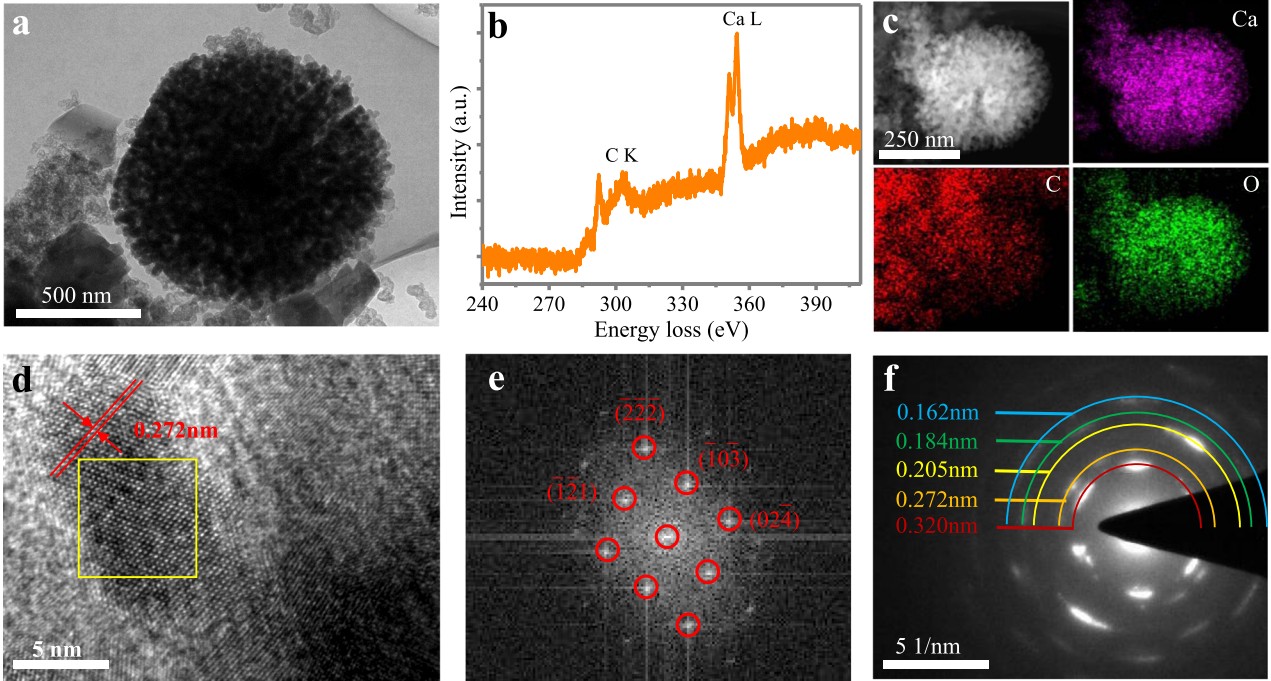

**Fig. 5 Ex situ TEM characterization of the Ca-inserted PT electrode. a** Low magnification TEM image. **b** EELS spectrum. **c** EDS mapping image of Ca, C and O. **d** high magnification TEM image. **e** The FFT image on the area shown in the yellow frame of (**d**) and (**f**) the SAED pattern.

obtain further insight into the exact site of Ca ion uptake (Fig. 6a). Generally, the more positive MESP (blue region) sites represent the nucleophilic center, whereas the more negative MESP areas (red region) prefer electrophilic reactions[50]. According to the MESP mapping results, a pronounced variation in electron distribution is observed between the carbonyl units and the aromatic rings of PT, which favors an electrostatic interaction between the molecules. The regions near the carbonyls of PT have more negative MESP values, suggesting that the carbonyls are more favorable for cation ion uptake, which shows good consistency with the XPS O1s and ex situ FTIR spectra[28]. According to the energy-band structure and density of states (Fig. 6b), the PT crystal is an indirect-gap material with a band gap of 1.77 eV, falling within the semiconductor range.

The diffusion of Ca ions is a primary task in the stacked and layered-like structure research field. The Ca migration pathway for a Ca atom confined within the PT crystal were investigated by the climb image nudged elastic band (CI-NEB) method (Fig. 6c), which contributes to finding the rules of calcium diffusion. The multimodal curve reflects the existence of several metastable states. The energy barriers for Ca diffusion and bond breaking/making were in the range of 0.18–0.35 eV as facilitated by the carbonyl units, and the migration path of Ca closely resembled an irregular spiral channel (Fig. 6c and Supplementary Movie 1). The cell with a PT:Ca ratio of 4:1 possesses an overall volume of ~0.3732 $nm^3$ after full relaxation (Fig. 6d), which is very close to the fresh PT cell volume (0.3703 $nm^3$), due to the efficient planar packing arrangement of the relaxed molecules.

According to Fig. 6c, four representative metastable states and their equilibrium configurations were further studied (Fig. 7). The inserted Ca ions tend to remain between the stacked layers of organic molecules, whereas at the interstitial space of the 1D channels (Fig. 7, Supplementary Fig. 15). Although the accommodation of one divalent $Ca^{2+}$ is expected to necessitate the enolation of two carbonyl groups, the π-π stacking nature of the crystal causes the negative charge from the enolates to be delocalized through the stacks. Therefore, each Ca ion tends to be

stabilized by four adjacent carbonyls to generate the optimal equilibrium configuration (the ground state configuration, State IV in Fig. 7b and Supplementary Fig. 15). The theoretical study on Ca transfer confirms that PT crystals with π-π stacked layers and 1D nanochannels show a proper scope of Ca-binding energies and easy ion transportation, allowing the fast storage of calcium ions in the nanochannels of PT crystals and contributing to the improvement of electrochemical performance.

To understand the performance of PT as a potential anode, a full aqueous CIB is necessary. We synthesized a high-voltage Prussian blue analogue (i.e., $KCoFe(CN)_6 \cdot xH_2O$) for the cathode material by a facile and simple coprecipitation route previously described[51]. The phase purity and crystal structure of the as-obtained $KCoFe(CN)_6 \cdot xH_2O$ are confirmed by XRD, which adopts a 3D cyanide-bridged architecture with a space group of Fm3m (Supplementary Fig. 16a). The as-prepared $KCoFe(CN)_6 \cdot xH_2O$ exhibits a single pair of well-defined redox peaks at 0.42/0.53 V vs. Ag/AgCl and one additional reduction peak at −0.13 V vs. Ag/AgCl in the first CV scan at 0.5 mV s⁻¹ (Supplementary Fig. 16b). Notably, the reduction peak at −0.13 V vs. Ag/AgCl is caused by the reduction in dissolved $O_2$ in the aqueous electrolyte and is only present in the first CV scan (Supplementary Fig. 17). The redox peaks at 0.42/0.53 V vs. Ag/AgCl are ascribed to the two redox couples of Co(III)/Co(II) and Fe(III)/Fe(II) in the as-prepared materials. The GCD profiles of the stabilized $KCoFe(CN)_6$ present one pair of stable operational voltage plateaus (Supplementary Fig. 16c), which is consistent with the CV plots. The rate performance is also explored, displaying specific capacities of 94.9, 80.7, 75.7, 69.9, and 66.8 mAh g⁻¹ at the specific currents of 0.2, 0.5, 1, 2, and 5 A g⁻¹, respectively. When the current is reduced to 1 A g⁻¹, the specific capacity immediately recovers with their reversal (Supplementary Fig. 16d). The cycling stability is examined at 2 A g⁻¹, delivering a capacity retention of 83.7% after 1000 cycles (Supplementary Fig. 16e). This achieved capacity and cycle ability far exceeds $Fe_4[Fe(CN)_6]_3$ and $KFe_{0.35}Mn_{0.65}Fe(CN)_6$ (59.5 mAh g⁻¹ and 60 mAh g⁻¹ at 1 A g⁻¹ (Supplementary Fig. 18), respectively, and

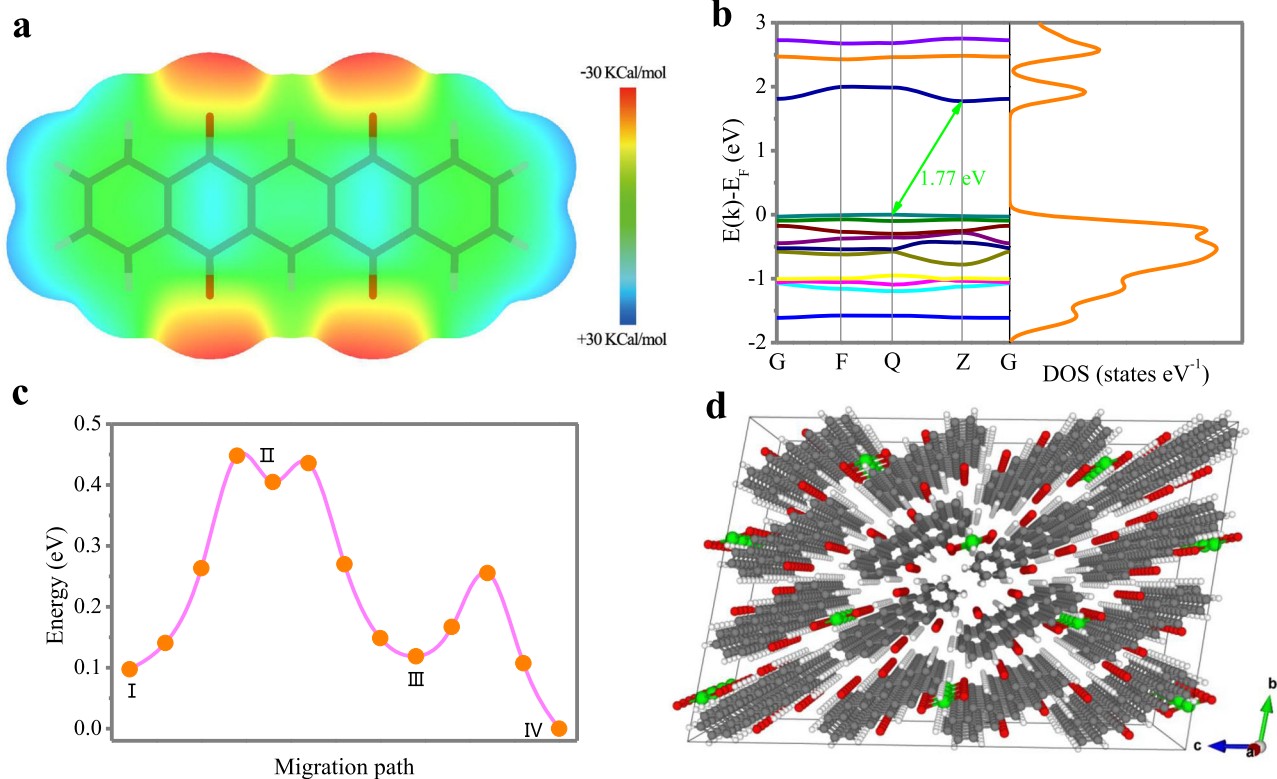

**Fig. 6 Theoretical calculations of the Ca-ion storage in a PT crystal. a** The MESP-mapped molecular van der Waals surface of PT. **b** the band structure and DOS of PT. **c** The diffusion pathway of a Ca ion confined within the channel of PT crystal. **d** Perspective view of a (4 × 4 × 10) Ca-inserted PT supercell with a PT: Ca concentration of 4:1.

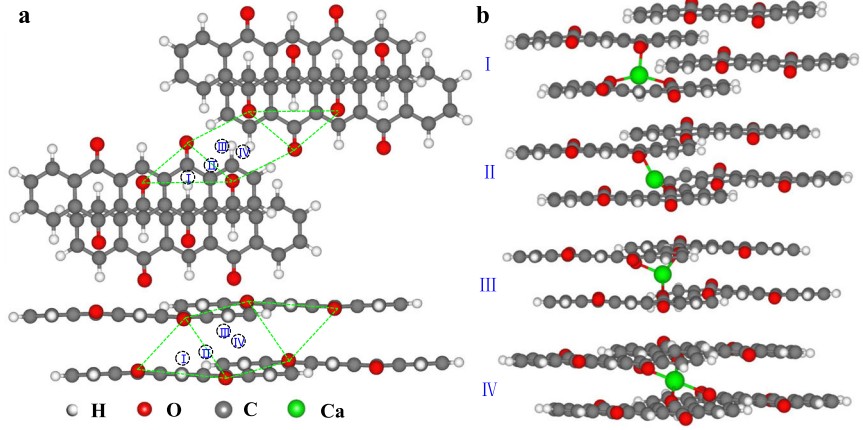

**Fig. 7 Spatial distribution of the Ca ions confined within the PT crystal channels. a** The top view. The dotted circle represents the relevant equilibrium position for Ca, the green dotted line shows the distance between the adjacent O atoms. **b** The front view.

that reported for CIBs cathode materials thus far (Supplementary Table 2), including $Ca_xCuHCF$ (42 mAh g$^{-1}$@12C)[32], KNiFe (CN)$_6$ (50 mAh g$^{-1}$@0.2C)[52], $K_xNiFe(CN)_6$ (50 mAh g$^{-1}$@25 μA cm$^{-2}$)[33], $NH_4V_4O_{10}$ (61 mAh g$^{-1}$@1 A g$^{-1}$)[53], $Mg_{0.25}V_2O_5 \cdot H_2O$ (70.2 mAh g$^{-1}$@0.1 A g$^{-1}$)[54], etc.

Considering the two described materials, we successfully assembled a full aqueous Ca-ion cell by utilizing $Ca^{2+}$ ion storage in the PT anode and insertion chemistry in the $KCoFe(CN)_6$ cathode (Fig. 8a and Supplementary Fig. 19). Considering the better rate capability of PT compared to $KCoFe(CN)_6$, a mass ratio of PT: $KCoFe(CN)_6 = 1:2.5$ is utilized. This full cell process can be concretely described by the individual CV curves of the

cathode and anode. As seen from Fig. 8b, the PT anode and PBA cathode invariably work in opposite potential directions, leading to a maximum cell voltage of 2.1 V. The typical CV profiles of the full cell produce three pairs of concessive reduction/oxidization peaks (Fig. 8c). The shape of the CV curves is well maintained with small polarization even at high scan rates of 100 mV s$^{-1}$. Then, a GCD test was performed on the full cell, generating specific capacities of 179.5, 129.6, 110.8, 99.6, 92.2, 82.5, 75.6 mAh g$^{-1}$ and 72.2 mAh $g_{PT}^{-1}$ at 2, 5, 10, 15, 20, 30, 40, and 50 A g$^{-1}$ based on the PT mass (Fig. 8d), corresponding to 51.3, 37.0, 31.7, 28.5, 26.3, 23.6, 21.6, and 20.6 mAh g$^{-1}$ based on the total mass of PT and $KCoFe(CN)_6$, respectively, proving good

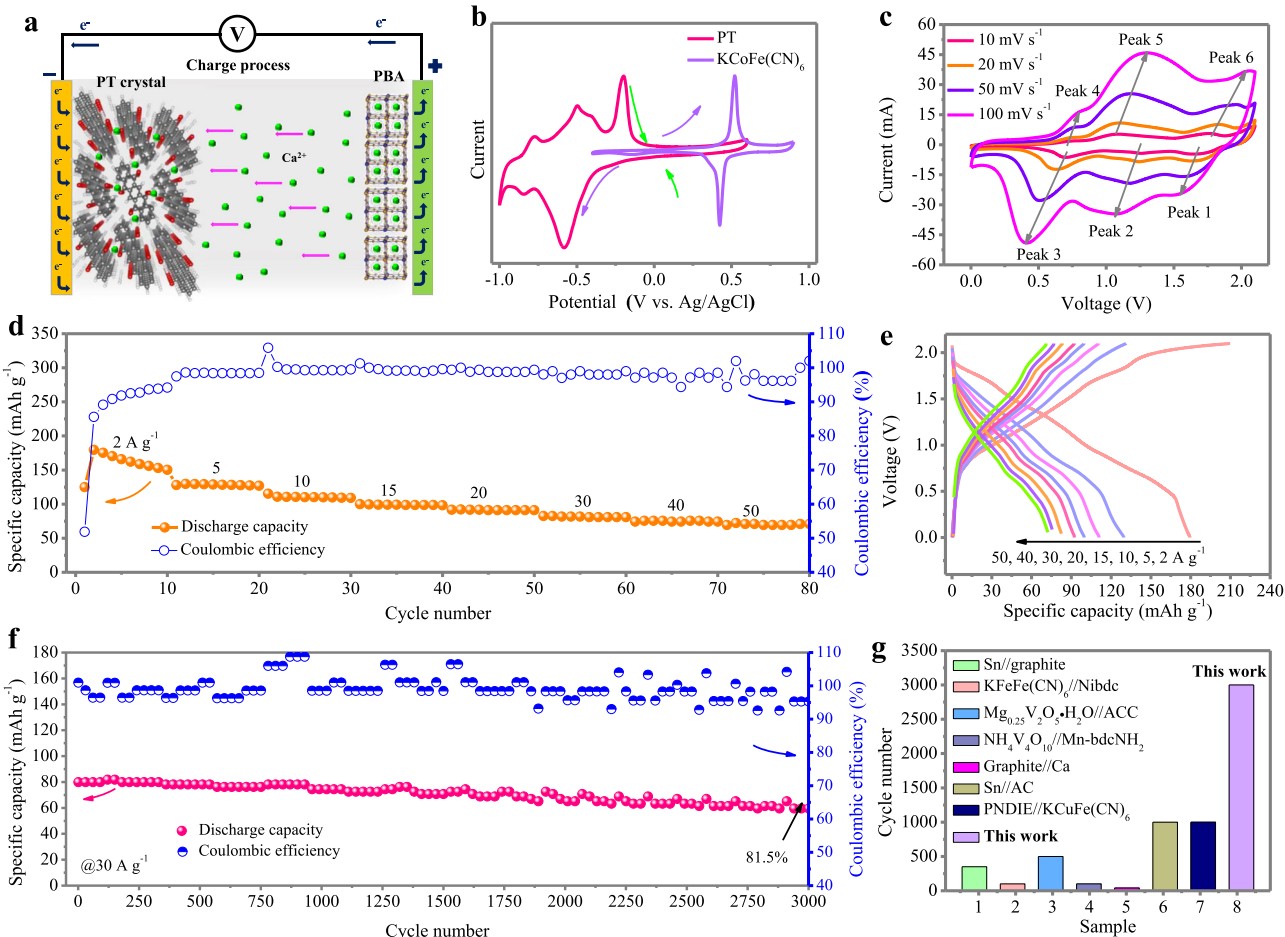

**Fig. 8 Full aqueous Ca-ion cell based on PT anode and KCoFe(CN)₆ PBA cathode. a** Schematic illustration of the reaction mechanism of the PT//PBA full cell. **b** Comparison of the CV curves of PT anode and PBA cathode. **c** The CV curves for the full cell at scan rates of 10–100 mV s⁻¹. **d** The rate capability at different specific currents. **e** the GCD curves at different specific currents. **f** Cycle performance at 30 A g⁻¹. **g** Cycle performance comparison with literature. See Supplementary Table 2 for reported data in literature.

rate capability. Notably, the relatively low Coulombic efficiency at 2 A g⁻¹ is caused mainly by the gradual activation of the PT anode; it rapidly increases to ~100% at subsequent specific currents. In addition, the operational voltage (2.1 V) and specific capacity are comparable to those of CIBs using an organic electrolyte, although aqueous batteries have a limited voltage window. The corresponding GCD curves are illustrated in Fig. 8e, revealing a sloping voltage profile with 0–2.1 V. The differential capacity (dQ/dV) curve exhibits distinct redox peaks at 2 A g⁻¹ (Supplementary Fig. 20). Correspondingly, a maximum specific energy of 200.4 Wh kg⁻¹ can be achieved at a specific power of 1861 W kg⁻¹ based on the mass of PT, which corresponds to 57.3 Wh kg⁻¹ based on the total mass of both PT and KCoFe(CN)₆. The self-discharge performance of the full cell was evaluated by standing for 24 h at 100% state of charge after the initial 20 cycles at 1 A g⁻¹, followed by discharging to 0 V. As shown in Supplementary Fig. 21, the capacity of the full cell decreased to 77% of its original discharge capacity after resting for 24 h and quickly recovered its original capacity in the subsequent cycles. The self-discharge behavior can be suppressed by electrode modification, electrolyte modulation and separator modification, etc[55–57]. Finally, charge–discharge cycling at a specific current of 30 A g⁻¹ was carried out, showing a capacity retention of ~81.5% after 3000 cycles (Fig. 8f), which is among the longest reported for CIB systems (both aqueous and nonaqueous) (Fig. 8g and

Supplementary Table 2), including Sn//graphite[18], Sn//Activated carbon (AC)[17], PNDIE//KCuFe(CN)₆[24], KFeFe(CN)₆//Ni-based metal–organic compound[44], NH₄V₄O₁₀//manganese 2-aminoterephthalate (Mn-bdcNH₂)[53], Mg₀.₂₅V₂O₅•H₂O//Activated carbon cloth (ACC)[54], graphite//Ca[19], etc.

In summary, the high charge density of divalent Ca²⁺ poses strong electrostatic interaction with the hosting lattice, resulting in the difficult Ca²⁺ storage. Moreover, the large ionic radius of Ca²⁺ (0.099 nm) leads to sluggish Ca²⁺ ion diffusion, making the reversible and fast storage of divalent Ca²⁺ highly challenging. We report an aromatic organic molecular crystal PT as a high-rate CIB anode. The long-range crystallinity and 1D diffusion channels considerably improve the Ca ion diffusion kinetics. Many benefits realized through these unique properties. The PT anode displayed a high specific capacity of 150.5 mA h g⁻¹ at 5 A g⁻¹, a capacity of 86.1 mA h g⁻¹ at a high current of 100 A g⁻¹, and stable cycling performance. A diffusion coefficient of Ca²⁺ ions in the PT anode is revealed by the GITT technique to be as high as 10⁻⁸–10⁻¹¹ cm² S⁻¹. Ex situ XPS, FTIR, and XRD collectively confirmed highly reversible and proton-assisted Ca²⁺ ion storage chemistry, accompanied by robust structural stability. Theoretical calculations suggest that the Ca ion migration path closely resembles an irregular spiral channel and that the most stable configuration is realized by the coordination of one Ca ion with four carbonyls in adjacent PT molecules. A full Ca-ion cell is

then capitalized with a KCoFe(CN)$_6$•xH$_2$O cathode, achieving a high voltage of 2.1 V and demonstrating the great promise of the as-developed PT as a potential anode material for aqueous ion batteries.

## Methods

**Electrode preparation.** 5,7,12,14-Pentacenetetrone (PT, 95% purity) was purchased from Macklin reagent Co., LTD. To prepare the PT anode, the PT material, acetylene black and polyvinylidene fluoride (PVDF) binder were mixed homogeneously in solids under stirring for 10 min in a weight ratio of 6:3:1. Then N-methyl-2-pyrrolidinone (NMP) solvent was added into the above mixture and stirred for 4 h to form a homogeneous slurry, which was then coated on a carbon cloth substrate (CeTech, Co., LTD) and dried under vacuum at 90 °C for 10 h. The carbon cloth substrate was treated in O$_2$ plasma for 10 min before use. The KCoFe(CN)$_6$·xH$_2$O cathode was prepared using the same slurry coating method except that a weight ratio of 6:2:2 was employed. The active material mass loading of the PT anode and the KCoFe(CN)$_6$·xH$_2$O cathode are ~1.5 mg cm$^{-2}$ and ~3.7 mg cm$^{-2}$, respectively.

**Cell fabrication.** Three-electrode beaker cells were first assembled with as-prepared PT or KCoFe(CN)$_6$ electrode (1 × 1 cm) as working electrode, Pt as the counter electrode and Ag/AgCl as the reference electrode (Supplementary Fig. 1). The distance between each electrodes are around 1 cm. The electrolyte employed was 8 mL aqueous CaCl$_2$ (99.9% metals basis, Aladdin) solution with different concentration (0.25 M, 0.5 M, 1 M, 2.2 M, and 3.4 M, respectively). Then, beaker-type full Ca-ion cells were assembled with PT as anodes and KCoFe(CN)$_6$ as cathodes in a mass ratio of 1:2.5 (anode/cathode) and 8 mL 1 M CaCl$_2$ aqueous solution as electrolyte (Supplementary Fig. 19).

**Electrochemical characterization.** Both CV experiment and GCD measurements was carried out on a CHI760E electrochemical workstation. GITT was performed using a battery testing system (LAND CT2001A). A series of current pulses of 200 mA g$^{-1}$ was applied on PT electrodes for 2 min followed by a 4 min relaxation process. The specific currents and specific capacities were calculated based on the mass of the active materials (i.e., PT or KCoFe(CN)$_6$·xH$_2$O, respectively). The specific energy (E, W h kg$^{-1}$) and corresponding specific power (P, W kg$^{-1}$) values of the full Ca-ion cell were calculated as follows:

$$E = \frac{\int IV(t)\mathrm{d}t}{3.6m} \tag{3}$$

$$P = \frac{3600E}{t} \tag{4}$$

where, I (A) is the applied current, V (V) is the voltage of the cell, t (s) is the corresponding discharge time, and m (g) is weight of electrode.

**Materials characterization.** The structure and phase composition were characterized by X-ray diffraction measurement (XRD, Rigaku D/max 2500/PC using CuKa radiation) and the diffraction data was collected at a step mode over the angular range of 5–60°. The microstructure and morphologies were characterized with field emission scanning electron microscopy (FE-SEM, HITACHI S4800, 10 kV, 5 mA) and transmission electron microscopy (TEM, TECNAI G2 F30, accelerating voltage of 200 kV) equipped with an energy-dispersive X-ray (EDX) spectroscopy (Oxford INCA) and electron energy-loss spectroscopy (EELS, Gatan). XPS analysis was measured in a PHI 5000 VersaProbe II equipped with Al Kα radiation and an Ar ion cluster sputtering gun. Attenuated Total Reflection Flourier transformed Infrared (ATR-FTIR) spectroscopy was conducted on a Thermo Scientific Nicolet iS 50 spectrometer. ICP-MS analysis was tested with Agilent ICP-MS 7700.

**Density functional theory calculations.** DFT calculations with the PBE functional were performed by using the Vienna ab initio Simulation Package[58,59]. The electron–ion interactions were presented by the frozen core all-electron projector augmented wave pseudopotentials, and generalized approximation of the electron exchange-correlation functional was adopted[60,61]. Periodic (3 × 2 × 1) supercell[28] was constructed to reveal the mechanism of calcium storage. All atoms were unfixed during the relaxation except the lattice parameters and basis vectors. The Kohn–Sham valence states were expanded in a plane-wave basis set with a kinetic cutoff energy of 500 eV, and geometries were optimized until the energy convergence criterion of 10$^{-4}$ eV and the maximum threshold force of 10$^{-2}$ eV Å$^{-1}$ were fulfilled. K-point sampling was performed by the Monkhorst-Pack scheme, with sampling grids of (3 × 3 × 3)[62]. To account for the weak dispersion interactions, these functionals were used in combination with the DFT-D2 correction[63]. The diffusion barrier of calcium ion in the channel of PT crystal was calculated based on the climb image nudged elastic band (CI-NEB) method[64]. A Monkhorst-Pack 3 × 3 × 3 mesh was used to sample the reciprocal space and the calculation

was considered to be converged when the residual force components on each atom were below 0.02 eV Å$^{-1}$.

**Reporting summary.** Further information on research design is available in the Nature Research Reporting Summary linked to this article.

## Data availability
The data that support the findings of this study are available from the corresponding author upon reasonable request.

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

## Acknowledgements
We acknowledge the financial support from Shenzhen Technical Plan Project (JCYJ20190808153609561), the National Key R&D Program of China under Project 2019YFA0705104, National Natural Science Foundation of China (22005207).

## Author contributions
Conceptualization: C.H., H.L., and C.Y.Z.; methodology: C.H., H.L., J.Z., and C.Y.Z.; investigation, C.H. and H.L.; computation: Y.L.; writing-original draft: C.H.; writing-review and editing, C.H., H.L. J.Z., Y.L., and C.Y.Z. H.L., and C.Y.Z. jointly supervised this work.

## Competing interests
The author declares no competing interests.
