## [Peer Review File · Nature Communications]

REVIEWER COMMENTS

Reviewer #1 (Remarks to the Author):

The authors report in this manuscript a version of a rechargeable aqueous Ca-ion battery using an aromatic organic molecular crystal, 5,7,12,14 Pentacenetetrone (PT) as an anode, and $\text{KCo}[\text{Fe}(\text{CN})_6]$ as a cathode in 1 M CaCl_2 electrolyte. The proposed proton-assisted uptake/removal of the Ca-ion during discharging\charging in the PT anode and also claim a supercapacitor like performance. The high current rate performance was supported by theoretical study and different characterization techniques. Finally, the authors proposed a full-cell with a maximum voltage of 2.1V and excellent cycling performance.

The reviewer is not convinced in many places and urged for more clarifications.

1. In the introduction, page 3, line #74, the authors compare the diffusion coefficient of the Ca^{2+} with Li^+ and Zn^{2+} in the different electrode materials. reference and the technique used for all the experiments should be indicated. The reviewer feels the idea of comparing the apparent diffusion coefficient is not correct here.

2. page 5, line# 132, the authors wrote that the PT anode exhibits two reduction peaks at -0.3 V and -0.66 V vs. $\text{Ag}|\text{AgCl}$ in 1 M CaCl_2 aqueous electrolyte. Since these two peaks are given in the SI figure S1, so it is suggested to incorporate the SI figure in the main article rather than mentioning Figure 2a.

3. Why the peaks at -0.3 V disappear and the -0.66 V peak shifts to -0.63 V upon cycling. Also, a shoulder peak at -0.85 V appears from the 2nd cycle..why?

4. Why in the first few cycles the Coulombic efficiency of the PT anode is greater than 100%?

5. In the GCD cycle of PT anode, at a current rate of 5 A g⁻¹, appears a small plateau (which is absent in higher current rate CD plots) at ~-1.0 V (Figure 2c). why?

6. PT anode showed intriguing capacity even at a very high current rate of 100 A g⁻¹. Does the crystal structure is maintained after cycling at 100 A g⁻¹ current rate? (post-cycling crystallographic information is essential here)

7. How much Ca^{2+} is taken up by the PT anode during discharge? Since the accommodation of one Ca^{2+} is difficult due to the large size and divalent nature of Ca-ion.

8. What is the theoretical specific capacity taken for $\text{KCo}[\text{Fe}(\text{CN})_6]$ cathode for calculating the C-rate? Is this the highest cathode material that can be used from the Prussian blue analog family in this study? what is the justification behind the use of this particular cathode?

10. The CV and CD profile of $\text{KCo}[\text{Fe}(\text{CN})_6]$ (mainly discharge plateau) is not consistent with each other...Why?

11. In the case of the cathode, why the peak at -0.22 V disappears with the subsequent cycle?

12. The CV profile of $\text{KCo}[\text{Fe}(\text{CN})_6]$ shows a hump at around 0.55 V. What corresponds to this peak?

13. From Figure 6d, the Coulombic efficiency is around 90% at the current rate of 2 A g⁻¹.please comment on what causes the loss of Coulombic efficiency?

14. Please provide self-discharge data of the full-cell and the what is the N/P ratio and discuss the strategy to stabilize the cell.

15. What is the basis of the calculation of the energy density value? Are these values based on merely the active mass of cathode (or anode or the total)?

16. Suggestion, the full-cell capacity of the battery should be calculated based on the total mass of cathode and anode.

Reviewer #2 (Remarks to the Author):

The manuscript by Han and collaborators investigated the intercalation of Ca ions into a novel organic cathode electrode, i.e., the 5,7,12,14 pentacenetrone. The manuscript is well written and touches upon an important topic –the identification of novel electrode chemistries as anode electrodes in Ca-based batteries, with significant Ca²⁺ diffusion coefficients of $\sim 10^{-8}$ – 10^{-11} S/cm. The investigation uses a number of techniques, including XRD, FT-IR, HSTEM, XPS, various electrochemistry techniques, and computation. In particular XPS of Ca and O are used to fingerprint at what voltage Ca and H react with the pentacenetrone. The intercalation of both H⁺ and Ca²⁺ is also verified via FT-IR measurements. Finally, an all-organic battery is prepared using the Prussian-blue like cathode KCoFe(CN)₆•xH₂O and the performance of the pentacenetrone are then tested. This manuscript deserves attention from Nature Communication after some revision are introduced.

1. The most pressing concern while reviewing this manuscript relates to the participation of H⁺ ions in the measured capacities. Indeed, the authors clarified that some peak in the cyclic voltammetry corresponds to the intercalation of protons in the electrode material. So what portion of the capacity claimed comes from proton insertion and Ca²⁺ intercalation? An CV of the material in absence of Ca²⁺ may clarify the net contribution towards the gravimetric capacity of protons?

Some other minor comments came out during the revision of the manuscript

2. Figure 1a is too small to be read and interpreted.

3. In both the abstract and the introduction is unclear that the authors are studying this novel material as a potential anode material. Authors should be more explicit.

4. In the introduction the author wrote the statement “The high charge density and relatively large ionic radius of divalent Ca²⁺ (0.99 Å) make them interact more strongly with the hosting lattice than monovalent cations, which results in sluggish Ca²⁺ ion diffusion in inorganic crystals.” It would be nice to see some of the seminal studies quantifying the sluggish mobility of on multivalent ions in material hosts cited, e.g. [10.1021/acs.chemmater.9b02692](https://doi.org/10.1021/acs.chemmater.9b02692) , [10.1021/acs.chemmater.5b02342](https://doi.org/10.1021/acs.chemmater.5b02342), [10.1038/s41467-017-01772-1](https://doi.org/10.1038/s41467-017-01772-1).

5. Furthermore the author may want to discuss the complexity of working directly with a Ca-metal anode, as discussed here [10.1039/D0EE02347](https://doi.org/10.1039/D0EE02347) , and thus the need of other alternative anode materials as pentacenetrone.

6. In the result section the authors confuse the diffusivity with ionic conductivity. While one would expect a linear relationship between the two, it’s not always the case. Please clarify this aspect.

7. Throughout the manuscript PH should be pH.

8. Figure 3h had poor resolution. Likewise Figure 3g could just look at the bands of interest; at the moment it’s very hard to identify any major spectral change.

Reviewer #3 (Remarks to the Author):

5,7,12,14 pentacenetetrone is by no means a new material. The material most likely exist with amorphous structure in the cycling (not the initial state, the XRD as shown in Fig 1 is the pristine state and during cycling, the structure will be subjected to change as it is not intercalation). That would make the 1D channel argument invalid because the ion transport mechanism is not intercalation, but adsorption.

Figure 4 seems to be a argument for supporting the intercalation, but since TEM is NOT a bulk technique and the crystalline structure is mostly some unreacted sphere. In the pairing with Prussian blue, the material acted more like a capacitor (ie, similar to a hard carbon).

Similarly, XPS and other technique are surface techniques, they do not reveal much about the bulk structure. Modeling is only useful when the hypothesis is correct.

Minor mistakes

1. This is

39 because the reversible plating/stripping of metallic calcium is only possible at elevated temperature

40 (50-100 °C) and in nonaqueous electrolyte⁸.

Actually Ca can be plated/stripped at RT and there is argument about the validity of the cited paper.

2.in Supp Info., ..."acetylene black and polyvinylidene fluoride (PVDF) 15 binder were mixed homogeneously"...

How was the mixing conducted?

Response to Reviewers' Comments

=====

Response to reviewers' comments

Reviewer #1:

1. In the introduction, page 3, line #74, the authors compare the diffusion coefficient of the Ca^{2+} with Li^+ and Zn^{2+} in the different electrode materials. Reference and the technique used for all the experiments should be indicated. The reviewer feels the idea of comparing the apparent diffusion coefficient is not correct here.

Reply: Thanks very much for your suggestion.

Following your suggestion, the comparison of apparent diffusion coefficient has been removed from the introduction section. In the main text, the references and techniques used for apparent diffusion coefficient measurement have been added as follows: The calculated diffusion coefficient of Ca^{2+} ions (D) in PT electrode during discharging and charging is in the range of $10^{-8}\sim 10^{-11} \text{ cm}^2 \text{ S}^{-1}$ by the GITT technique (**Figure 2i**), which is significantly higher than that of Li^+ in LiFePO_4 ($10^{-15}\sim 10^{-17} \text{ cm}^2 \text{ S}^{-1}$ determined by GITT) (*Solid State Ionics*, 2002, 148: 45-51) and $\text{Li}_4\text{Ti}_5\text{O}_{12}$ ($10^{-11}\sim 10^{-12} \text{ cm}^2 \text{ S}^{-1}$ determined by potentiostatic intermittent titration technique, $10^{-12}\sim 10^{-13} \text{ cm}^2 \text{ S}^{-1}$ determined by CV) (*J. Solid State Chem.*, 2004, 177: 2094-2100; *Energy Environ. Sci.*, 2012, 5, 9595-9602), Zn^{2+} in $\alpha\text{-MnO}_2$ ($10^{-15}\sim 10^{-17} \text{ cm}^2 \text{ S}^{-1}$ determined by GITT) (*Chem. Commun.*, 2015, 51: 9265-9268), Ca^{2+} in other materials (e.g., Ca^{2+} in Nibdc, $5.3\times 10^{-14} \text{ cm}^2 \text{ S}^{-1}$ determined by electrochemical impedance spectroscopy) (*ACS Sustain. Chem. Eng.*, 2020, 8: 2596-2601), etc.

We have revised the manuscript accordingly. Please see the highlighted part on page 10.

2. page 5, line# 132, the authors wrote that the PT anode exhibits two reduction peaks at -0.3 V and -0.66 V vs. Ag|AgCl in 1 M CaCl_2 aqueous electrolyte. Since these two peaks are given in the SI figure S1, so it is suggested to incorporate the SI figure in the main article rather than mentioning Figure 2a.

Reply: Thanks very much for your suggestion.

Following your suggestion, we have incorporated Figure S1 into the main text. Please see Figure 2a on page 8.

3. Why the peaks at -0.3 V disappear and the -0.66 V peak shifts to -0.63 V upon cycling. Also, a shoulder peak at -0.85 V appears from the 2nd cycle. why?

Reply: Thanks very much for your question.

(1) In a degassed CaCl_2 electrolyte, there is no cathodic peak at -0.3 V, suggesting that the cathodic peak at -0.3 V in the untreated electrolyte is ascribed to the reduction of dissolved O_2 . This small cathodic peak at -0.3 V only presents in the first cathodic scan and will disappear in subsequent cycles. Similar phenomena were also observed in other aqueous battery systems, such as aqueous lithium ion batteries (*Energy Fuels*, 2013, 27, 2, 1162-1167).

(2) There is an activation process for the PT anode, which can be verified by the gradually enhanced current density in CV profiles (Figure 2a), the increased specific capacity at 5 A g^{-1} (Figure 2b), as well as the capacity increase at initial cycles during the cycling test (Figure 2g). This activation process is most possibly related to the morphological evolution of large PT bulks to small nanowires (Figure S11 and S12), which not only leads to a gradual exposure of the inner active surface, but also contributes to fast ion diffusion and efficient charge transfer due to better accessibility for charge carriers infiltrating into PT materials. Consequently, a slight peak shifts from -0.66 V to -0.63 V and the gradual appearance of the shoulder peak at -0.85V is observed. The activation process of electrode materials is commonly observed in both organic and inorganic electrode materials, including disodium rhodizonate (*Nature Energy*, 2017, 2, 861-868), diquinoxalino [2,3-a:2',3'-c] phenazine (HATN) (*Angew. Chem. Int. Ed.* 2020, 59, 4920-4924), Poly(2,6-Anthraquinonyl Sulfide) (*Adv. Energy Mater.* 2020, DOI:10.1002/aenm.202002780), MoS_2 (*Energy Storage Mater.*, 2019, 19, 94-101), Ni and 1,4-benzenedicarboxylate based metal organic compound (denoted as Nibdc) (*ACS Sustainable Chem. Eng.* 2020, 8, 7, 2596-2601), etc.

We have revised the manuscript accordingly. Please see the highlighted part on page 5.

4. Why in the first few cycles the Coulombic efficiency of the PT anode is greater than 100%?

Reply: Thanks very much for your comments.

A gradual activation process is observed for the PT anode. At the beginning of the GCD test, the activation process is not completed, and the current is not fully applied on the PT electrode material, which leads to some side reactions like hydrogen evolution reaction (HER) and is mainly responsible for the high Coulombic efficiencies in the first few cycles. This HER process appears as a small plateau at the tail end of discharge. With the increase of cycling numbers, the PT electrode material was gradually activated, the Coulombic efficiency, therefore, was stabilized to ~100%. This can be collectively verified by the gradually increased specific capacities during the first few cycles (Figure 2b). Similar phenomena were also observed in other aqueous battery systems, such as pyrene-4,5,9,10-tetraone (PTO)//MnO₂ battery (*Nat. Commun.*, 2020, 11, 959), (NH₄)_{1.47}Ni[Fe(CN)₆]_{0.88}//3,4,9,10-perylenetetracarboxylic diimide (PTCDI) ammonium-ion battery (*Angew. Chem. Int. Ed.* 2017, 56, 13026-13030), δ -V₂O₅//Zn batteries (*ACS Energy Lett.*, 2020, DOI:10.1021/acsenergylett.0c01767), NiCo₂S_{4-x}//Zn battery (*Nano Energy*, 2020, 77, 105165), VOPO₄-based magnesium ion battery (*Nano Lett.*, 2018, 18, 6441-6448), etc.

We are sorry for having not made this point clear. For better clarity, we have revised the manuscript accordingly. Please see the highlighted part on page 7.

5. In the GCD cycle of PT anode, at a current rate of 5 A g⁻¹, appears a small plateau (which is absent in higher current rate CD plots) at ~-1.0 V (Figure 2c). why?

Reply: Thanks very much for your comments.

As just discussed above, at the beginning of the activation process, the current is not fully applied on the PT electrode material, which leads to side reactions like the hydrogen evolution reaction (HER). This HER process appears as a small plateau at

the tail end of discharge. After completing the activation, the HER process can be fully restrained. Therefore, this small plateau is no longer observable.

We have added related discussion into the revised manuscript and revised Figure 2c with the 10th GCD curve at each current rate. Please see the highlighted part on page 7.

6. PT anode showed intriguing capacity even at a very high current rate of 100 A g⁻¹. Does the crystal structure is maintained after cycling at 100 A g⁻¹ current rate? (post-cycling crystallographic information is essential here)

Reply: Thanks very much for your comments.

Following your suggestion, the XRD profile of PT electrode after cycling at 100 A g⁻¹ for 100 cycles is measured and compared with the fresh PT electrode (**Figure S9**). It is seen that the original peaks of PT are well maintained, except with some intensity decrease, indicating the robust structural stability of PT materials during extremely large current densities.

We have revised the manuscript accordingly. Please see the highlighted part on page 13 in the revised manuscript and Figure S9 in the supporting information.

Figure S9. XRD profile of PT electrode after cycling at 100 A g⁻¹ for 100 cycles and its comparison with the fresh PT electrode.

7. How much Ca^{2+} is taken up by the PT anode during discharge? Since the accommodation of one Ca^{2+} is difficult due to the large size and divalent nature of Ca-ion.

Reply: Thanks very much for your comments.

The PT anode store charge carriers *via* an ‘ion-coordination’ mechanism where the cations coordinate to the negatively charged oxygen atoms upon electrochemical reduction of the carbonyl groups, and uncoordinate reversibly during the reverse oxidation. This ‘ion-coordination’ reaction is generally not restricted by the choice of charge carrier ions, which is capable of storing both monovalent ions and multivalent metal ions.

Since PT contains only light and atmosphere rich elements (i.e., C, O, H), it is difficult to precisely confirm the amount of Ca^{2+} taken by PT anode. Nevertheless, we used both inductively coupled plasma-mass spectrometry (ICP-MS) analysis and XPS analysis to quantify the content of Ca. Firstly, the concentration of Ca in the whole PT electrode is measured to be 16.9 g kg^{-1} by the ICP-MS technique (**Table S3**); Secondly, the atomic ratio of Ca in the surface of the discharged PT electrode is ~ 6.0 at.% by XPS analysis.

We have revised the manuscript accordingly. Please see the highlighted part on page 12, page 14 in the revised manuscript and Table S3 in the supporting information.

8. What is the theoretical specific capacity taken for $\text{KCo}[\text{Fe}(\text{CN})_6]$ cathode for calculating the C-rate? Is this the highest cathode material that can be used from the Prussian blue analog family in this study? What is the justification behind the use of this particular cathode?

Reply: Your comments are sincerely appreciated.

(1) The theoretical specific capacity of electrode materials can be calculated by the following equation (*Chem*, 2018, 4, 1-28):

$$Q = \frac{nF}{3.6Mw}$$

Where Q is the specific capacity (mAh g^{-1}), n represents the number of transferred electrons per molecule, F is the Faraday constant (C mol^{-1}), and the M_w is the molecule weight (g mol^{-1}). Therefore, the theoretical specific capacity of KCoFe(CN)_6 cathode is calculated to be 172.9 mAh g^{-1} based on a two-electron transfer reaction. However, we used the current density (i.e., A g^{-1}) instead of C-rate to characterize the rate performance of KCoFe(CN)_6 cathode, for example, 0.2, 0.5, 1, 2, 5 A g^{-1} .

(2) When prescreening for suitable cathode candidates, we prepared three different Prussian blue analog materials including the KCoFe(CN)_6 , $\text{Fe}_4[\text{Fe(CN)}_6]_3$, $\text{KFe}_{0.35}\text{Mn}_{0.65}\text{Fe(CN)}_6$, and compared their electrochemical performance. It is found that KCoFe(CN)_6 exhibited an enhanced specific capacity up to 94.9 mAh g^{-1} at 0.2 A g^{-1} , which is significantly larger than that of $\text{Fe}_4[\text{Fe(CN)}_6]_3$ (59.5 mAh g^{-1} at 1 A g^{-1}), KCuFe(CN)_6 (60 mAh g^{-1} at 1 A g^{-1} , **Figure S17**) and literature reported KNiFe(CN)_6 ($50 \text{ mAh g}^{-1}@0.2\text{C}$) (*Nano Lett.*, 2013, 13: 5748-5752), Ca_xCuHCF ($42 \text{ mAh g}^{-1}@12\text{C}$) (*ACS Appl. Mater. Interfaces*, 2020, 12: 11489-11503), $\text{K}_x\text{NiFe(CN)}_6$ ($50 \text{ mAh g}^{-1}@25 \mu\text{A cm}^{-2}$) (*Electrochim. Acta* 2016, 207: 22-27), $\text{Mg}_{0.25}\text{V}_2\text{O}_5 \cdot \text{H}_2\text{O}$ ($70.2 \text{ mAh g}^{-1}@0.1 \text{ A g}^{-1}$) (*ACS Energy Lett.*, 2019, 4: 1328-1335), $\text{NH}_4\text{V}_4\text{O}_{10}$ ($61 \text{ mAh g}^{-1}@1 \text{ A g}^{-1}$) (*J. Mater. Chem. A*, 2018, 6: 22645-22654), etc. This is because there are two redox reactions of Co(II)/Co(III) and Fe(II)/Fe(III) couples in KCoFe(CN)_6 , in which both Co and Fe ions can effectively contribute to the two-electron energy storage process, resulting in an enhancement in the specific capacity. Therefore, KCoFe(CN)_6 was selected as the cathode active material due to its high specific capacity.

We have revised the manuscript accordingly. Please see the highlighted part on page 17 in the revised manuscript and Figure S17 in the supporting information.

10. The CV and CD profile of $\text{KCo[Fe(CN)}_6]$ (mainly discharge plateau) is not consistent with each other...Why?

Reply: Thanks very much for your comments.

The potential difference between the CV profiles and the GCD curves is mainly caused by the large scan rate of CV measurement (i.e., 10 mV s^{-1}). The CV profile

recorded at a reduced scan rate of 0.5 mV s^{-1} presents a pair of well-distinguished redox peaks at 0.42/0.53 V vs. Ag/AgCl, respectively, which coincides perfectly with the GCD profiles.

We have revised the manuscript accordingly for better clarity. Please see page 16 in the revised manuscript and Figure S15b in the supporting information.

11. In the case of the cathode, why the peak at -0.22 V disappears with the subsequent cycle?

Reply: Thanks very much for your comments.

The cathodic peak at -0.22 V @ 10 mV s^{-1} (now is -0.13 V @ 0.5 mV s^{-1}) is ascribed to the reduction of dissolved O_2 in the electrolyte and only present in the first cathodic scan, as revealed by the comparison with the CV profile measured in a degassed CaCl_2 electrolyte (**Figure S16**). The same phenomena were also observed in PT anode as discussed in question 3.

We have revised the manuscript accordingly. Please see page 16 in the revised manuscript and Figure S16 in the supporting information.

12. The CV profile of $\text{KCoFe}(\text{CN})_6$ shows a hump at around 0.55 V. What corresponds to this peak?

Reply: Thanks very much for your comments.

Since the $\text{KCoFe}(\text{CN})_6$ offers two active couples of Co(III)/Co(II) and Fe(III)/Fe(II), it will require two steps for Ca^{2+} intercalation/deintercalation. The two reduction peaks appear as a pair of redox peaks and a hump at a high scan rate.

We have revised the manuscript accordingly for better clarity. Please see page 16 in the revised manuscript and Figure S15b in the supporting information.

13. From Figure 6d, the Coulombic efficiency is around 90% at the current rate of 2 A g^{-1} . Please comment on what causes the loss of Coulombic efficiency?

Reply: Thanks very much for your comments.

As discussed in question 4&5, the PT electrode (working electrode) exhibits a Coulombic efficiency of >100 % when tested at a low current density in the three electrode configuration, mainly due to the gradual activation process of PT anode. Consequently, when the PT electrode was used as the anode of the full battery, the as-assembled full cell exhibits a Coulombic efficiency of around 90% at an initial current rate of 2 A g⁻¹. It rapidly increases to ~100% at following current densities.

We have added more discussion in the revised manuscript for better clarity. Please see the highlighted text on page 18 in the revised manuscript.

14. Please provide self-discharge data of the full-cell and what is the N/P ratio and discuss the strategy to stabilize the cell.

Reply: Thanks very much for your question.

The full cell was assembled with PT as anode and KCoFe(CN)₆ as cathode in a mass ratio of 1:2.5 (anode/cathode). The self-discharge performance of the full cell was evaluated by standing for 24 h at 100% state of charging (SOC) after the initial 20 cycles at 1 A g⁻¹, followed by discharging to 0 V. As uncovered in newly added Figure S19, the capacity of the full cell decreased to 77% of its original discharge capacity after 24 hours and quickly recovers its original capacity in the subsequent cycles.

The strategies to stabilize the cell and inhibit the self-discharge can be summarized as follows (*ACS Nano* 2020, 14, 4916–4924; *Phys. Chem. Chem. Phys.*, 2016, 18, 661-680; *J. Electrochem. Soc.*, 2015, 162, A5047-A5053):

(1) Modifying the electrodes. Tuning the self-discharge rate can be achieved by adjusting the surface chemistry and particle size of the PBA cathode and the PT anode. The effects of surface chemistry modification on self-discharge are very important in suppressing the self-discharge process.

(2) Modulating the electrolyte. Polymer hydrogel electrolytes can effectively hinder side reactions such as the shuttle effect by adulterants and hydrogen evolution reaction (HER)/oxygen evolution reaction (OER), which largely suppress the self-discharge behavior of aqueous CIBs. Besides, the utilization of electrolyte

additives is another facile and economically effective way of alleviating the self-discharge process.

(3) Tuning the separator. It was found that the separator with the negative or positive charges can control the ionic movement, thus the corresponding shuttle effects were inhibited. As a result, tuning the separator is a useful strategy to suppress the self-discharge. Furthermore, using ion-exchange separators is also beneficial to trap impurities and minimize Faradaic side reactions, thereby suppressing self-discharge in aqueous CIBs.

We have added more discussion in the revised manuscript. Please see the highlighted text on page 18 and page 19 in the revised manuscript.

15. What is the basis of the calculation of the energy density value? Are these values based on merely the active mass of cathode (or anode or the total)?

Reply: Thanks very much for your question.

The energy density is calculated according to equations 1 in the supporting information:

$$E = \frac{I \int V(t) dt}{m} \quad (S1)$$

Where I is the discharge current (A), t is the corresponding discharge time (s), $V(t)$ denotes the cell voltage (V), m refers to the weight (g). Since galvanostatic charge-discharge was used for the performance measurement, the energy density therefore can be calculated through integrating the discharge profiles (i.e., the area under the discharge profile).

(1) When calculated based on the mass of PT anode, the energy density is 200.4 Wh kg^{-1} by integrating the area under the discharge profile at 2 A $\text{g}_{\text{PT}}^{-1}$.

$$E = \frac{I \int V(t) dt}{m} = 200.4 \text{ Wh } \text{kg}^{-1}$$

(2) When calculated based on the total mass of both PT anode and $\text{KCoFe}(\text{CN})_6$ cathode (with a mass ratio of 1:2.5), the energy density is 57.3 Wh kg^{-1} at 2 A $\text{g}_{\text{PT}}^{-1}$.

$$E = \frac{200.4 \text{ Wh kg}^{-1} * m_{PT}}{(1 + 2.5) * m_{PT}} = 57.3 \text{ Wh kg}^{-1}$$

We have added both energy densities in the revised manuscript. Please see page 18 in the revised manuscript.

16. Suggestion, the full-cell capacity of the battery should be calculated based on the total mass of cathode and anode.

Reply: Thanks very much for your suggestion.

Following your suggestion, the full-cell capacity has been calculated based on the total mass of $\text{KCoFe}(\text{CN})_6$ cathode and PT anode. Please see the highlighted text on Page 18 in the revised manuscript.

Reviewer #2:

1. The most pressing concern while reviewing this manuscript relates to the participation of H^+ ions in the measured capacities. Indeed, the authors clarified that some peak in the cyclic voltammetry corresponds to the intercalation of protons in the electrode material. So what portion of the capacity claimed comes from proton insertion and Ca^{2+} intercalation? An CV of the material in absence of Ca^{2+} may clarify the net contribution towards the gravimetric capacity of protons?

Reply: Thanks very much for your valuable comments.

Following your suggestion, we have measured the CV and GCPL profile of PT in 1M HCl electrolyte.

(1) In the HCl electrolyte, the PT electrode shows two reduction peaks at 0.06 V, 0 V vs. Ag/AgCl and two oxidization peaks at 0.05 V, 0.12 V vs. Ag/AgCl, respectively (**Figure 3a**), which is related to the proton uptake/ release with the carbonyl groups of PT. Corresponding CV curves in a pH=7.8 solution (the same pH value with 1 M CaCl_2 aqueous solution) can be obtained by shifting the CV curves measured in 1M HCl according to the Nernst equation (see the Supporting Information for details).(*Adv Mater* 2020, 32(16): 2000338) It is noted that the location of the two

oxidization peaks after shifting partially overlapped with the oxidization peaks at -0.47 V and -0.38 V vs. Ag/AgCl measured in 1M CaCl₂ electrolyte, suggesting that these two peaks at -0.47 V and -0.38 V correspond to the proton storage process in the PT electrode. Therefore, by comparing the under area of the two oxidization peaks with the total anodic scan, it is calculated that ~49 % of the total capacity comes from the contribution of proton uptake.

(2) We also measured the GCD profile of PT in 1M HCl. PT electrode delivers a specific discharge capacity of 64.6 mAh g⁻¹ at 5 A g⁻¹ (**Figure S6**), which corresponds to ~43 % of the 150.5 mAh g⁻¹ measured in 1M CaCl₂. This coincides well with the CV measurement. Therefore, it is estimated that around 40~50 % of the total capacity comes from the contribution of proton uptake.

We have added the above discussion into the revised manuscript. Please see page 11 in the revised manuscript.

Some other minor comments came out during the revision of the manuscript

2. Figure 1a is too small to be read and interpreted.

Reply: Thanks very much for your comments.

Following your suggestion, Figure 1a (now is Figure 1b) has been enlarged for better interpretation. We have revised the manuscript accordingly. Please see page 6, Figure 1b.

3. In both the abstract and the introduction is unclear that the authors are studying this novel material as a potential anode material. Authors should be more explicit.

Reply: Thanks very much for your comments.

Following your suggestion, “PT electrode” has been changed into “PT anode” throughout the revised manuscript to better illustrate the potential application of PT materials.

4. In the introduction the author wrote the statement “The high charge density and relatively large ionic radius of divalent Ca²⁺ (0.99 Å) make them interact more

strongly with the hosting lattice than monovalent cations, which results in sluggish Ca^{2+} ion diffusion in inorganic crystals.” It would be nice to see some of the seminal studies quantifying the sluggish mobility of on multivalent ions in material hosts cited, e.g. [10.1021/acs.chemmater.9b02692](https://doi.org/10.1021/acs.chemmater.9b02692), [10.1021/acs.chemmater.5b02342](https://doi.org/10.1021/acs.chemmater.5b02342), [10.1038/s41467-017-01772-1](https://doi.org/10.1038/s41467-017-01772-1).

Reply: Thanks very much for your comments.

These references are highly related to our manuscript, and we have added more discussion regarding the sluggish mobility of multivalent ions in material hosts:

“The high charge density and relatively large ionic radius of divalent Ca^{2+} (0.99 Å) make them interact more strongly with the hosting lattice than monovalent cations, which results in sluggish Ca^{2+} ion diffusion in inorganic crystals (*Chem. Mater.*, 2019, 31(19): 8087-8099). For example, Ceder and co-workers investigated the migration of several multivalent ions (Mg^{2+} , Zn^{2+} , Ca^{2+} , and Al^{3+}) in spinel Mn_2O_4 , olivine FePO_4 , layered NiO_2 , and orthorhombic $\delta\text{-V}_2\text{O}_5$ and found that the mobility of multivalent ions is consistently lower than Li^+ , and the barrier of different bivalent ions depends very strongly on the hosting structure (*Chem. Mater.*, 2015, 27(17): 6016-6021). High multivalent ions mobility in solids is only possible by judicious tuning of crystal structure and chemistry (*Nat. Commun.*, 2017, 8(1): 1759).”

Please see the highlighted text on page 2 and Ref 14, 15 and 16 in the revised manuscript.

5. Furthermore the author may want to discuss the complexity of working directly with a Ca-metal anode, as discussed here [10.1039/D0EE02347G](https://doi.org/10.1039/D0EE02347G), and thus the need of other alternative anode materials as pentacenetetrone.

Reply: Thanks very much for your suggestion.

We have added more discussion on the complexity of working directly with a Ca-metal anode and incorporated some of the seminal studies quantifying this point, including, *Nat. Mater.*, 2018, 17, 16-20, *ACS Energy Lett.* 2019, 4, 9, 2271–2276, *Energy Environ. Sci.*, 2019, 12, 3496-3501, *Energy Environ. Sci.*, 2020, 13, 3423-3431.

Please see the highlighted text on page 2 in the revised manuscript.

6. In the result section the authors confuse the diffusivity with ionic conductivity. While one would expect a linear relationship between the two, it's not always the case. Please clarify this aspect.

Reply: We regret for this oversight and thank you very much for pointing it out.

The diffusion coefficient measured by the GITT technique refers to the diffusion of ions within the electrode material during battery reaction, which is different from the ionic conductivity of electrolytes. We have double-checked the description throughout the manuscript and revised our manuscript accordingly.

Please see the highlighted part on page 10 in the revised manuscript.

7. Throughout the manuscript PH should be pH.

Reply: We regret for this oversight and thank you very much for pointing it out.

“PH” has been changed into “pH” throughout the manuscript.

8. Figure 3h had poor resolution. Likewise Figure 3g could just look at the bands of interest; at the moment it's very hard to identify any major spectral change.

Reply: Thanks for your comments.

Following your suggestions, high-resolution images have been given in the revised manuscript. Moreover, magnified FTIR and XRD images showing the major changes have also been added. Please see the revised Figure 3g and 3h on Page 12 in the revised manuscript.

Reviewer #3:

1. 5,7,12,14 pentacenetrone is by no means a new material.

Reply: Thanks very much for reviewing our manuscript and your comments are sincerely appreciated.

We agree with the reviewer that pentacenetrone is not a new material in materials science. However, this does not affect the novelty of our work based on the following reasons:

Firstly, in the field of multivalent ion based batteries, the reversible and high-capacity storage of multivalent ions remains a grand challenge due to the high charge density and large ionic radius of multivalent ions. Though it has been reported in other fields of materials science, **PT, as an aromatic organic molecules crystal, has not been explored in multivalent ion batteries, including Ca ion batteries.** By comparing with the state-of-the-art inorganic electrodes for Ca ion batteries, such as PBA, layered oxides, etc., PT demonstrates remarkably superior electrochemical performance in terms of specific capacity ($150.5 \text{ mA h g}^{-1}$ at 5 A g^{-1}), rate capability (86.1 mA h g^{-1} at an extremely high current of 100 A g^{-1} (315 C)) and cycle stability (3000 cycles). **This manuscript opens new opportunities to explore aromatic organic electrode materials for Ca-ion storage and other multivalent ion battery applications (Zn^{2+} , Mg^{2+} , Al^{3+} , etc.), but not limited to PT.**

Secondly, to date, little understanding has been achieved on the storage chemistry and structural evolution of aromatic organic materials during ion storage, despite it is of vital significance to search for high-performance organic electrodes. In this manuscript, we reported the highly reversible coordination and uncoordination of both protons and Ca^{2+} with carbonyl groups of PT anodes and **firstly clarified the important role of protons in the storage chemistry of Ca^{2+} ions.** More importantly, **the findings of this work can be extended to understand the charge storage characteristics of other aromatic organic molecular crystals,** such as phenanthrenequinone (9,10-PQ) (**Figure S5**), perylene-3,4,9,10-tetracarboxylic dianhydride (PTCDA).

We have added more discussions to illustrate the novelty of this work and extended the findings of this work to a broad context of aromatic organic crystals. Therefore, we believed this manuscript has demonstrated significant novelty and importance in the field of energy storage.

We have revised the manuscript accordingly. Please see the highlighted part on page 1, page 3, page 4, page 6, page 11, etc.

2. The material most likely exist with amorphous structure in the cycling (not the initial state, the XRD as shown in Fig 1 is the pristine state and during cycling, the structure will be subjected to change as it is not intercalation). That would make the 1D channel argument invalid because the ion transport mechanism is not intercalation, but adsorption.

Reply: Thanks very much for your comments.

(1) To investigate the structural change of PT anode during, XRD measurement was conducted on PT anodes that are subjected to different cycles including 200, 500, 1000, 1500, 2000 cycles at 30 A g^{-1} (**Figure S10**). It can be seen that it still demonstrated good crystallinity throughout the cycling test, indicating the robust structural stability of PT materials during repeated cycling. So does the PT anode that is cycled at a small current density of 0.1 A g^{-1} (**Figure S8**) and an extremely large current density of 100 A g^{-1} (**Figure S9**).

(2) Regarding the ion storage mechanism, we agree with the reviewer that it is not the conventional intercalation mechanism observed in inorganic electrodes. Organic carbonyls store charge via an ‘ion-coordination’ mechanism where the cations coordinate to the negatively charged oxygen atoms upon electrochemical reduction of the carbonyl groups, and uncoordinate reversibly during the reverse oxidation. This enolation reaction of quinone ($-\text{C}=\text{O}$) to quinone salts ($-\text{C}-\text{O}-\text{M}$) can be regarded as a kind of ‘chemical adsorption’, leading to well-defined CV peaks and noticeable charge/discharge plateaus. Moreover, compared with traditional intercalation/deintercalation based energy storage mechanism, this kind of ‘chemical adsorption’ reaction is beneficial for sustaining high structural stability of PT materials during repeated cycling.

We have revised the manuscript accordingly. Please see page 13 in the revised manuscript and Figure S8, S9, and S10 in the supporting information.

3. Figure 4 seems to be an argument for supporting the intercalation, but since TEM is NOT a bulk technique and the crystalline structure is mostly some unreacted sphere.

Reply: Thanks very much for your comments.

Indeed, as the reviewer said, TEM is micro-scale sensitive and there is some unreacted PT. However, low magnification SEM images in **Figure S11 and S12** clearly revealed the formation of large amounts of crystalline spheres as the discharging products, which are absent in the initial PT samples. These crystalline spheres consist of Ca, O, and C elements as revealed by EELS analysis. Moreover, the formation and disappearance of these rice-like spheres are highly reversible during the discharge and charge process (**Figure S11**). For these unreacted PT materials, they show bulky morphology beneath these newly formed spheres and they are Ca-free as revealed by **Figure S13**, which is quite different from the reacted PT.

To further confirm the bulk storage of Ca^{2+} , in-depth XPS profiles were performed to examine the spatial difference by Ar ion cluster sputtering. As shown in **Figure S7**, after etching for 10 nm and 20 nm, both of these discharged PT electrodes still exhibit featured in-depth XPS peaks of Ca2p, which clearly verify the bulk storage of Ca in PT anodes. Furthermore, we also used inductively coupled plasma-mass spectrometry (**ICP-MS**) analysis to quantify the content of Ca in the whole PT anode, which is measured to be 16.9 g kg^{-1} (**Table S3**).

We have added high magnification SEM images, In-depth XPS and ICP-MS data of the discharged PT anodes in the revised manuscript. Please see page 13-14 in the revised manuscript, and Figure S7, S11, S12, S13 and Table S3 in the supporting information.

4. In the pairing with Prussian blue, the material acted more like a capacitor (ie, similar to a hard carbon).

Reply: Thanks very much for your comments.

Truly as the reviewer said, this PT material exhibits a supercapacitor-level high rate capability, which is also explicated in our manuscript. Actually, with the rapid progress achieved in nanoscience and nanotechnologies, the boundary between

battery material and capacitive material (particularly pseudocapacitive materials) has been becoming blurred in recent years. There is difficulty and debate on how to accurately distinguish these two types of materials (*ACS Nano* 2018, 12, 3, 2081-2083; *Small*, 2020, 16 (37), 2002806; *Energy Environ. Mater.*, 2019, 2 (1), 30-37). Typically, capacitive materials basically exhibit near-rectangular CV profiles and almost linear GCD curves, while battery-type materials show obvious CV peaks and detectable charge/discharge plateaus in GCD profiles. When pairing with Prussian blue, the GCD profile of the full battery presents obvious discharge plateaus at around 1.7V, 1.3V and 0.75V at 2 A g^{-1} , despite these discharge plateaus become indistinct at high rates. Moreover, the differential capacity (dQ/dV) curve of the full battery also exhibits distinct redox peaks at 2 A g^{-1} (**Figure S18**). So, the PT material is considered as a battery-type anode material despite its capacitor-level high rate capability.

We have added related discussion into the revised manuscript. Please see the highlighted part on page 18 in the revised manuscript and Figure S18 in the supporting information.

5. Similarly, XPS and other technique are surface techniques, they do not reveal much about the bulk structure. Modeling is only useful when the hypothesis is correct.

Reply: Thanks very much for your comments.

We agree with the reviewer that XPS is a surface-sensitive technique. Therefore, in-depth XPS profiles were performed to examine the spatial difference by Ar ion cluster sputtering. As shown in **Figure S7**, after etching for 10 nm and 20 nm, both of these discharged PT electrodes still exhibit featured in-depth XPS peaks of Ca2p, which clearly verify the bulk storage of Ca in PT anodes. Furthermore, we also used inductively coupled plasma-mass spectrometry (ICP-MS) analysis to quantify the content of Ca in the whole PT anode, which is measured to be 16.9 g kg^{-1} (Table S3).

Our modeling is based on the above experimental analysis and therefore we believe it is reasonable and correct. We have added the above discussion into the revised manuscript. Please see the highlighted part on page 13, page 14 in the revised

manuscript and Figure S7, Table S3 in the supporting information.

Minor mistakes

1. This is because the reversible plating/stripping of metallic calcium is only possible at elevated temperature (50-100 °C) and in nonaqueous electrolyte⁸. Actually Ca can be plated/stripped at RT and there is argument about the validity of the cited paper.

Reply: We regret for this oversight and thank you very much for pointing it out.

We have revised the description into “This is because the reversible plating/stripping of metallic calcium is only possible in judiciously tailored nonaqueous electrolyte”. We have incorporated some of the seminal studies quantifying this point, including, *Nat. Mater.*, 2018, 17, 16-20, *ACS Energy Lett.* 2019, 4, 9, 2271–2276, *Energy Environ. Sci.*, 2019, 12, 3496-3501, *Energy Environ. Sci.*, 2020, 13, 3423-3431.

Please see the highlighted text on page 2 in the revised manuscript.

2. in Supp Info., ..."acetylene black and polyvinylidene fluoride (PVDF) 15 binder were mixed homogeneously"...How was the mixing conducted?

Reply: Thanks very much for your comments.

The active material, acetylene black (conductive additive) and polyvinylidene fluoride (PVDF) binder were mixed homogeneously in solid powders under stirring for 10 min in a weight ratio of 6 : 3 : 1. Then N-methyl-2-pyrrolidinone (NMP) solvent was added into the above mixture and stirred for 4 h to form a homogeneous slurry, which was then coated on a carbon cloth substrate (CeTech, Co., LTD).

We have added more details regarding the electrode preparation process. Please see the highlighted text on Page S1 in the supporting information.

REVIEWER COMMENTS

Reviewer #1 (Remarks to the Author):

Overall the response to the reviewer is fair and scientific. There is a minor mistakes in Figure S9, it should be written as after 100 cycles not "100 mA/g". Suggesting for a minor revision before publishing.

Reviewer #2 (Remarks to the Author):

The authors have diligently addressed all my doubts.

Reviewer #3 (Remarks to the Author):

The concept of this paper is to create a full batteries by using both the Prussian blue cathode/organic anode, is not going to help the area in general.

Intercalation of Ca^{2+} in Prussian blue is well documented, and mechanism at the anode side is most like going through a conversion/supercapacitor mechanism.

I felt overwhelmed by the unfocused characterization, and the conclusion is not impressive. (See previous comments about TEM and XPS, they are not the correct method to characterize bulk structure change).

After revision, the 'ion coordination' is suggested as mechanism. As there is very little discussed about ion coordination, and most of organics electrodes behave in a similar fashion, so the work lack of originality and impact.

The concept paper is also confusing: it try pair two electrodes that both are not well adapted in academia or industry for batteries. Multivalent ion (Ca^{2+} or Mg^{2+}) have been shown to work in both electrodes independently, and combining them does not make this paper very new in ideas or science more advanced.

Response to Reviewers' Comments

Dear Reviewers:

Thank you very much for reviewing our manuscript entitled "Proton-assisted Ultrafast Calcium Ion Storage in Aromatic Organic Molecular Crystal with Coplanar Stacked Structure and 1D Molecular Tunnels" (Manuscript ID: NCOMMS-20-39815A). These comments are all valuable and very helpful for improving the quality of our manuscript. We have studied these comments carefully and revised our manuscript accordingly. All comments have been replied as below and revisions are highlighted in our revised manuscript with a yellow background.

=====

Response to reviewers' comments

Reviewer #1:

Overall the response to the reviewer is fair and scientific. There is a minor mistakes in Figure S9, it should be written as after 100 cycles not "100 mA/g". Suggesting for a minor revision before publishing

Reply: Thanks very much for your positive comments.

The mistake in Figure S9 has been revised. Also, following your suggestion, we have polished our manuscript by native English speakers from Nature Language Editing Service, aiming at eliminating the grammatical, typos and linguistic errors. Based on the polished version, we have further double-checked our manuscript thoroughly to ensure clear and accurate expression.

Please check the new caption of Figure S9.

Reviewer #2:

The authors have diligently addressed all my doubts.

Reply: Thanks very much for your positive comments.

Reviewer #3:

1. The concept of this paper is to create a full batteries by using both the Prussian blue cathode/organic anode, is not going to help the area in general.

Reply: Thank you for the comment. Actually, our paper focus on a high-rate and long-life Ca-metal-free anode material instead of a full battery. The full battery is to demonstrate the application of the anode material we developed. The high-performance anode materials for Ca ion batteries and their working mechanism remain challenging in the past years.

Current research on CIBs is still in its infancy due to the complexity of working directly with a Ca-metal anode and the lack of high-performance anode materials. To date, only a handful of anode materials have been demonstrated, and their performances, especially the specific capacity and cycling stability (<1,000 cycles), are still far from satisfactory. Therefore, the exploration of Ca-metal-free anode candidates capable of reversible Ca^{2+} storage is critically important yet challenging.

The aromatic organic molecules crystal family has remained unexplored for CIBs. Besides, little understanding has been gained on the storage chemistry and the structural evolution of organic materials during multivalent ion storage, including Ca^{2+} storage.

This work first identified that aromatic organic molecular crystal, represented by pentacenetetrone (PT), **can serve as an extremely high-rate and long-life Ca-metal-free anode material, which has been an obstacle for the development of CIBs for years.** PT demonstrates remarkably superior electrochemical performance than state-of-the-art inorganic materials in terms of specific capacity ($150.5 \text{ mA h g}^{-1}$ at 5 A g^{-1}), rate capability (86.1 mA h g^{-1} at an extremely high current of 100 A g^{-1} (315 C)) and cycle stability (3000 cycles). More importantly, the Ca^{2+} ion storage chemistry and structural evolution of aromatic organic materials were systematically studied, which first identified a highly reversible and proton-assisted storage chemistry of Ca^{2+} ions with the carbonyl groups of PT anode in aqueous CIBs.

The findings of this work constitute a major advance in developing high-performance Ca-metal-free anode materials and represent a good starting point towards high-rate and long-life CIBs.

Other main highlights of this manuscript are as follows:

(1) The loosely π - π stacked layered structure and the presence of rich 1D tunnels in aromatic organic PT crystals render PT an exceptionally high-performance anode material for CIBs. **It opens new opportunities to explore novel organic crystals as high-performance metal-free anode materials for CIBs and other multivalent ion battery applications** (Zn^{2+} , Mg^{2+} , Al^{3+} , etc.).

(2) Little understanding has been achieved on the storage chemistry and structural evolution of aromatic organic materials during ion storage, despite it is of vital significance to search for high-performance organic electrodes. In this manuscript, **we reported the highly reversible adsorption and desorption of Ca^{2+} with carbonyl groups of PT anodes. More importantly, the findings of this work can be extended to understand the charge storage characteristics of other aromatic organic molecular crystals**, such as phenanthrenequinone (9,10-PQ), perylene-3,4,9,10-tetracarboxylic dianhydride (PTCDA).

(3) **The role of protons is of key importance yet undiscernible in aqueous CIBs.** Here, a highly reversible and proton-assisted storage chemistry of Ca^{2+} ions with the carbonyl groups of PT anode is firstly identified in aqueous CIBs, which are accompanied by robust structural stability. **The above findings shed light on the charge storage mechanisms of Ca ions in aqueous CIBs and other aqueous batteries.**

2. Intercalation of Ca^{2+} in Prussian blue is well documented, and mechanism at the anode side is most like going through a conversion/supercapacitor mechanism.

Reply: Thanks for your valuable comments.

We agree with that the PT anode does not follow the conventional intercalation mechanism observed in inorganic electrodes. The enolation reaction of PT can be regarded as a kind of 'chemical adsorption', leading to well-defined CV peaks and noticeable charge/discharge plateaus. Following the reviewer's and the editor's suggestion, we have changed the word "intercalation" with "confined within" and adopted the terms "chemical adsorption" and "chemical desorption" to better describe the charge storage mechanism.

3. I felt overwhelmed by the unfocused characterization, and the conclusion is not impressive. (See previous comments about TEM and XPS, they are not the correct method to characterize bulk structure change).

Reply: Thanks very much for your comments.

We have used multiple techniques including **XRD, SEM, TEM, EELS, in-depth XPS, FTIR, ICP-MS** and various **electrochemical characterization methods (CV, GCPL, GITT)** to collectively investigate the phase, structure, valence, composition and morphology evolution of PT during the discharge/charge processes. The testing results are briefly categorized as follows:

(1) **FTIR spectra** revealed the conversion of carbonyl groups ($\text{C}=\text{O}$, 1668 cm^{-1}) to enolate groups ($\text{C}-\text{O}^-$, $\sim 1460\text{ cm}^{-1}$);

(2) **XPS** explorations strongly evidenced the reversible uptake of Ca^{2+} during discharging ($355\text{-}340\text{ eV}$) and its removal after recharging. Meanwhile, XPS O 1s spectra revealed the appearance of the C–O peak (530.3 eV) after discharging and its disappearance after charging.

(3) **XRD** results exhibit well-retained characteristic peaks, indicating the robust structural stability of PT materials during repeated cycling.

(4) **SEM and TEM** images clearly revealed the formation of large amounts of spheroidal particles as the discharging products, which are absent in the initial PT samples. These crystalline spheres consist of Ca, O, and C elements as revealed by EELS mapping analysis. For these unreacted PT materials, they show bulky morphology beneath these newly formed spheres and they are Ca-free, which is quite different from the reacted PT.

(5) **In-depth XPS** profiles were performed to examine the spatial difference by Ar ion cluster sputtering. After etching for 10 nm and 20 nm, the PT anode still exhibits featured in-depth XPS peaks of Ca2p, which clearly verify the storage of Ca in PT anodes.

(6) **ICP-MS** analysis is to quantify the content of Ca in the whole discharged PT anode, which is measured to be 16.9 g kg^{-1} , further support our assumption.

In summary, the above characterization results collectively support the ‘chemical adsorption’ mechanistic assumption where the Ca^{2+} cations adsorb to the negatively charged

oxygen atoms upon electrochemical reduction of the carbonyl groups, and desorb reversibly during the reverse oxidation.

We have added a new section to summarize and categorize the characterization techniques and their conclusion in the Supporting Information. Please see the highlighted text on Page S2.

4. After revision, the 'ion coordination' is suggested as mechanism. As there is very little discussed about ion coordination, and most of organics electrodes behave in a similar fashion, so the work lack of originality and impact.

Reply: Thanks very much for your comments.

(1) The lack of high-performance anode material is a main obstacle for the development of Ca ion batteries. This work developed a high-performance anode material for Ca ion batteries.

(2) The accommodation of Ca^{2+} in PT anode is quite complicated because of the divalent nature of Ca^{2+} , particularly in aqueous CIBs, which is not well documented in literature. This is because the accommodation of one Ca^{2+} is expected to necessitate the enolation of at least two carbonyl groups. This is very different from monovalent ions (e.g., Li^+ , Na^+ , or K^+), where only one carbonyl group binds one cation. Our work suggests that Ca ions tend to be stored between the stacked layers of organic molecules, whereas at the interstitial space in the 1D tunnels and are stabilized by four adjacent carbonyls in adjacent PT molecules, which hasn't been reported before.

(3) Thirdly, the role of protons during the storage of Ca^{2+} remains undiscernible in aqueous CIBs. Here, a highly reversible and proton-assisted storage chemistry of Ca^{2+} ions with the carbonyl groups of PT anode is firstly identified in aqueous CIBs, which are accompanied by robust structural stability.

Therefore, the above findings have shown significant originality by shedding light on the role of protons and the charge storage mechanisms of Ca^{2+} ions in organic materials for aqueous CIBs.

More discussions regarding the charge storage mechanism have been added in the revised manuscript. Please see the highlighted text on page 10 and page 15 in the revised main text.

5. The concept paper is also confusing: it try pair two electrodes that both are not well adapted in academia or industry for batteries. Multivalent ion (Ca^{2+} or Mg^{2+}) have been shown to work in both electrodes independently, and combining them does not make this paper very new in ideas or science more advanced.

Reply: Thanks very much for your comments.

(1) The development of CIBs is still in its infancy stage. Particularly, there is great difficulty in working directly with a Ca-metal anode, which has been an obstacle for CIBs for years. **The main focus of this work is to develop novel Ca-metal-free anode using organic pentacenetetrone (PT), aiming at tackling the challenges of CIB anode, rather than pairing two electrodes to create a full battery.**

(2) This work first identified that pentacenetetrone (PT), one aromatic organic molecular crystal, can serve as an extremely high-rate and long-life Ca-metal-free anode material. The PT demonstrated remarkably superior electrochemical performance than state-of-the-art inorganic materials in terms of specific capacity ($150.5 \text{ mA h g}^{-1}$ at 5 A g^{-1}), rate capability (86.1 mA h g^{-1} at 100 A g^{-1} ($\sim 315 \text{ C}$)) and cycle stability (3000 cycles), providing an excellent anode candidate for CIBs. The above finding opens a new window to develop organic crystals as high-rate and long-life CIB anodes and encourages the exploration of more novel alternative anodes.

(3) More importantly, the Ca^{2+} storage chemistry and the structural evolution of anode in aqueous CIBs haven't been studied. In this work, **we first identified a highly reversible and proton-assisted storage mechanism of Ca^{2+} ions with the carbonyl groups of PT anode. The findings have shown significant originality by shedding light on the vital role of protons**, the structural evolution and the Ca^{2+} ions storage mechanism of aqueous CIBs, which haven't been reported before. It represents a good starting point to understand the charge storage chemistry of Ca ions in organic electrode materials and reaction mechanism of aqueous CIBs.

(4) To demonstrate the application of PT anode in a full battery, a Prussian blue cathode is paired with PT. The as-assembled full battery confirms that PT works very well in full CIB, proving that it is a good anode candidate for aqueous CIBs.

Therefore, the key point of this work is to tackle the challenge of Ca metal anode by developing an excellent Ca-metal-free anode candidate (i.e., PT), rather than creating a full battery using the Prussian blue cathode/organic anode. Our work is expected to pave the way towards the development of organic-based anode materials for Ca ion batteries with superior rate capability and long lifespan.